# Optineurin modulates the maturation of dendritic cells to regulate autoimmunity through JAK2-STAT3 signaling

Jiajia Wang [1,3], Jiaying Wang[1,3], Wenxiang Hong[1,3], Lulu Zhang[1], Liqian Song[1], Qi Shi[1], Yanfei Shao[2], Guifeng Hao[2], Chunyan Fang[1], Yueping Qiu[1], Lijun Yang[1], Zhaoxu Yang[1], Jincheng Wang [1], Ji Cao[1], Bo Yang[1], Qiaojun He[1] & Qinjie Weng [1✉]

Optineurin (OPTN) has important functions in diverse biological processes and diseases, but its effect on dendritic cell (DC) differentiation and functionality remains elusive. Here we show that OPTN is upregulated in human and mouse DC maturation, and that deletion of *Optn* in mice via *CD11c*-Cre attenuates DC maturation and impairs the priming of CD4[+] T cells, thus ameliorating autoimmune symptoms such as experimental autoimmune encephalomyelitis (EAE). Mechanistically, OPTN binds to the JH1 domain of JAK2 and inhibits JAK2 dimerization and phosphorylation, thereby preventing JAK2-STAT3 interaction and inhibiting STAT3 phosphorylation to suppress downstream transcription of IL-10. Without such a negative regulation, *Optn*-deficient DCs eventually induce an IL-10/JAK2/STAT3/IL-10 positive feedback loop to suppress DC maturation. Finally, the natural product, Saikosaponin D, is identified as an OPTN inhibitor, effectively inhibiting the immune-stimulatory function of DCs and the disease progression of EAE in mice. Our findings thus highlight a pivotal function of OPTN for the regulation of DC functions and autoimmune disorders.

[1] Center for Drug Safety Evaluation and Research, Zhejiang Province Key Laboratory of Anti-Cancer Drug Research, College of Pharmaceutical Sciences, Zhejiang University, Hangzhou 310058, China. [2] Department of Rheumatology, Zhejiang Provincial People's Hospital, People's Hospital of Hangzhou Medical College, Hangzhou, Zhejiang 310014, China. [3]These authors contributed equally: Jiajia Wang, Jiaying Wang, Wenxiang Hong. ✉email: wengqinjie@zju.edu.cn

Dendritic cells (DCs) are the most potent antigen-presenting cells (APCs) in mammalian immune systems, playing a crucial role in the development and maintenance of immune responses and tolerance. In steady state, immature DCs (iDCs) undergo partial or homeostatic maturation, characterized by low surface expression of co-stimulatory molecules, such as CD80, CD86, and CD40, and increased anti-inflammatory cytokines secretion, which is important for maintaining peripheral immune tolerance by inducing T cells anergy and promoting regulatory T cells (Treg cells) production[1,2]. During inflammation or exposure to 'danger signals' (such as toll-like receptor signaling), DCs are stimulated for maturation to trigger T-cell immune activation. Mature DCs (mDCs) can migrate from peripheral tissue to draining lymph nodes (DLN), where they can present peptide of antigens by high expressed major histocompatibility complex (MHC) molecules and co-stimulatory molecules for full T-cell activation[1,3]. Besides, DCs can be generally divided into two major subsets, conventional DCs (cDCs) and plasmacytoid DCs (pDCs)[4]. The cDCs are classical antigen-presenting cells, while the pDCs are the main producers of type I IFNs further serving innate immunity.

Therefore, many efforts have been made to exploit the potential of DC therapies, ranging from the suppression of autoimmunity, the establishment of transplant tolerance, to the induction of tumor immunity[5,6]. The adoptive transfer of immature DCs (iDCs) emerged as a new strategy to restore immune tolerance, with a therapeutic implication in treating autoimmune diseases. The iDCs generated in vitro inhibited the experimental autoimmune encephalitis (EAE) in mice[7]. However, successful pursuit of these applications requires fully understanding the factors regulating DC differentiation and function[1,3]. Identifying the critical factors that control DC maturation is crucial for a better understanding of the immune response system and the discovery of novel clinical interventions for T cells-mediated autoimmune diseases.

Optineurin (OPTN) is a multifunctional protein associated with various neuronal diseases[8]. Recently, OPTN has attracted considerable attention for being identified differentially expressed or mutant in several immune disorders, including Crohn's disease (CD) and Paget's disease of bone[9,10]. The well-characterized functions of OPTN are selective autophagy through its LC3 interaction region and ubiquitin binding that depends on its ubiquitin (Ub) binding domain (UBD)[11]. OPTN controls immunity mainly through nuclear factor-kappaB (NF-κB) and interferon (IFN) signaling pathways, which are also related to ubiquitination. It was shown to act as a negative regulator of tumor necrosis factor (TNF) induced NF-κB activation by competing with NF-κB essential modulator (NEMO) for ubiquitinated receptor-interacting protein (RIP)[12]. And the production of type I IFNs in the host defense against infections required the interaction between ubiquitin chains and OPTN[13]. However, it remains elusive whether OPTN plays a specific role during DC maturation and consequently mediates the immune system.

Herein, we report that depletion of *Optn* in mice by *CD11c*-Cre promotes JAK2-STAT3 phosphorylation and the forming of IL-10/JAK2/STAT3/IL-10 positive feedback loop, thus impairing DC maturation and the priming of CD4+ T cells to prevent the development of autoimmunity in mice. Moreover, the natural product, Saikosaponin D (SSD), attenuates EAE symptom by inhibiting OPTN expression, suggesting that interfering OPTN could be beneficial for the therapeutic treatment of autoimmune diseases.

## Results

**OPTN is required for DC maturation**. To explore the expression pattern of OPTN during DC maturation or activation, human monocyte-derived dendritic cells (MoDCs) were stimulated with bacterial lipopolysaccharide (LPS), a toll-like receptor 4 (TLR4) agonist, for maturation[14]. Array analysis showed that *Optn* gene expression was significantly increased in mature MoDCs, similar as the dramatically upregulated DC maturation-related genes, such as *CD40*, *TNF*, *IL-1B*, *CXCL9*, and *CXCL10* (Fig. 1a). Next, we cultured murine bone marrow-derived dendritic cells (BMDCs) by GM-CSF (50 ng/mL) and interleukin (IL)-4 (20 ng/mL), and then acquired mature BMDCs by LPS stimulation. Be different from the common GM-CSF (20 ng/ml) and IL-4 (5 ng/ml) system that comprised both of DCs and macrophages[15], our system acquired quite a low number of CD11c+MHC-II-CD11b^high cells and almost no CD115+ macrophages (Supplementary Fig. 1), thus representing an effective BMDCs culture system in vitro. As expected, qRT-PCR verified the elevated expression of *Optn* in mature BMDCs (Fig. 1b). Immunostaining and western blotting analysis also indicated that OPTN was accumulated in CD11c+ human MoDCs and BMDCs after LPS or PolyI:C treatment (Fig. 1c), suggesting that OPTN may be involved in the regulation of DC maturation or activation.

To understand the biological role of OPTN in DC maturation, we generated CD11c+ cells conditional *Optn* knockout mice (*Optn*^fl/fl; *CD11c*-Cre mice, hereafter called *Optn*^ΔCD11c mice) and confirmed the total deletion of OPTN in BMDCs by qRT-PCR and western blotting analysis (Fig. 1d). The number of CD11c+ DCs from bone marrow and spleen were comparable between *Optn*^ΔCD11c mice and control mice (Fig. 1e), and the expression levels of classic co-stimulatory molecules CD80 and CD86 were also changed faintly in *Optn* deficient BMDCs (Fig. 1f), suggesting that OPTN was dispensable for the generation of DCs. Interestingly, the phenotypic maturation of LPS challenged *Optn* deficient BMDCs was inhibited obviously compared to that of controls (Fig. 1g). These data indicate that *Optn* deficiency blocks DC maturation.

**OPTN regulates the T cells priming function of DC**. It is well known that mature DCs can secrete anti-inflammatory cytokines to complete immunogenicity. Our results showed that *Optn* deficient BMDCs secreted more anti-inflammatory cytokines like IL-10 and transforming growth factor (TGF)-β, although failed to inhibit the production of pro-inflammatory cytokines like IL-6 and IL-12 (Fig. 2a). Encouraged by these results, we further found that cDCs were decreased, while pDCs were increased in *Optn*^ΔCD11c mice (Fig. 2b). Meanwhile, the percentages of peripheral tissue-derived migratory DCs (M-DCs) but not resident DCs (R-DCs) were overtly reduced in the DLN of *Optn*^ΔCD11c mice (Fig. 2c). These data represent a potential role of OPTN in regulating the immunogenic phenotype of DCs, which may further control the T-cell activation.

Given that the function of DCs switches from capturing and processing antigens to predominantly presenting them to T cells after maturation[16], we next examined if OPTN ablation in DCs affects the ability of T-cell priming in vitro. Flow cytometry results showed that *Optn*-deficient BMDCs had a much better ability than control cells to ingest more fluorescein isothiocyanate-dextran (FITC-dextran), suggesting that *Optn* deficiency in DCs improved its ability of antigen uptake (Fig. 2d). Then, we applied an in vitro T-cell priming model which involved the activation of OT-II CD4+ T cells co-incubated with DCs pulsed with a specific peptide, chicken OVA$_{323-339}$[17]. As expected, *Optn*-deficient DCs inhibited the proliferation of OT-II CD4+ T cells, as measured by CFSE staining (Fig. 2e). In addition, *Optn*-deficient DCs exhibited an impaired ability to initiate the production of autoimmune factors IFN-γ and IL-17 in OT-II CD4+ T cells (Fig. 2f). Thus, these results suggest that *Optn* deletion suppresses the T-cell priming function of DCs in vitro.

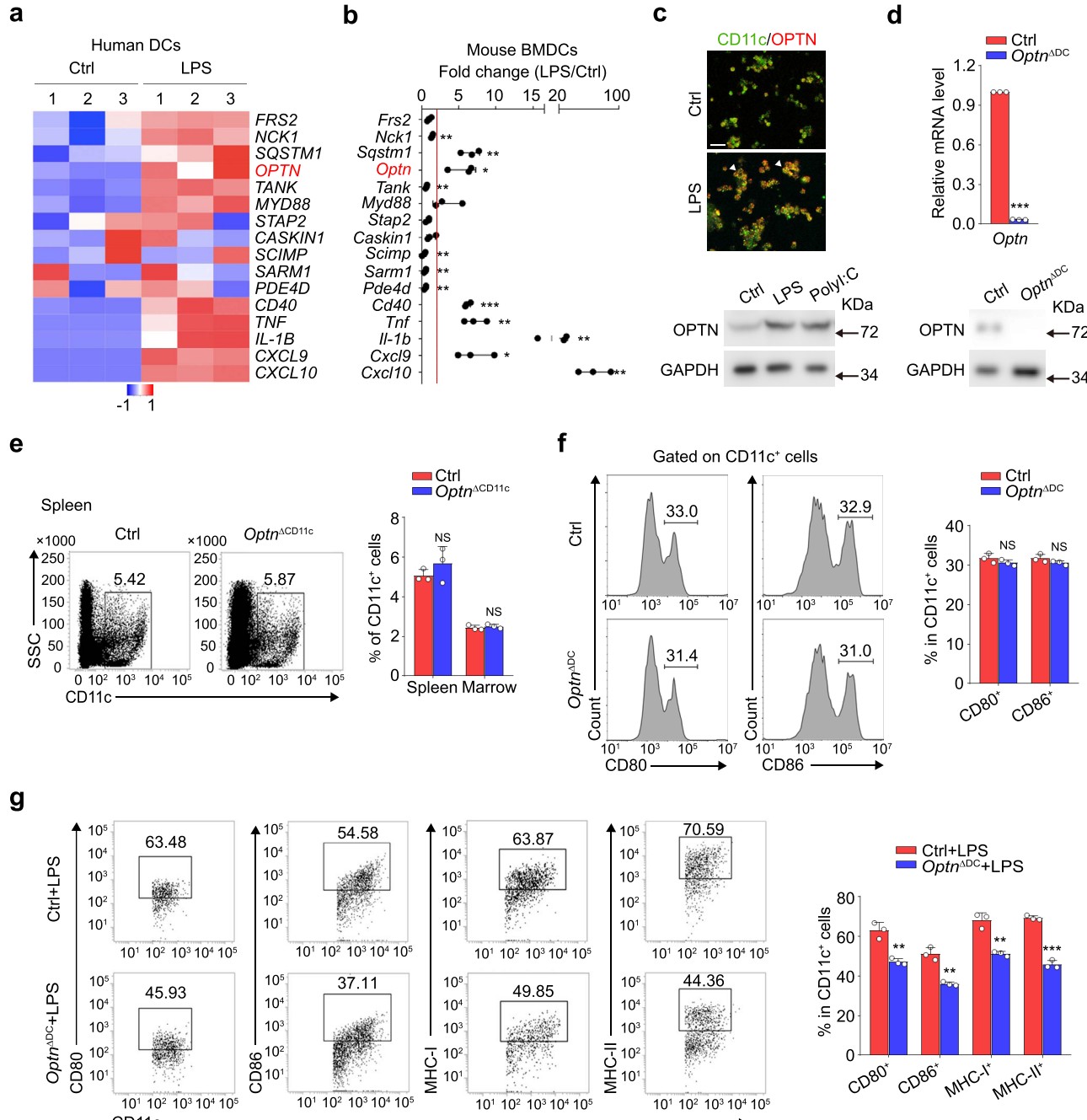

**Fig. 1 OPTN expression is required for DC maturation. a** Heatmap of the indicated gene expression levels in human MoDCs stimulated with LPS for 8 h or unstimulated from GEO dataset (GSE2706). **b** qRT-PCR analysis of selected gene expressions in BMDCs stimulated with LPS for 8 h. Red line, relative fold change =2. $n = 3$ independent experiments, $P_{(Nck1)} = 0.0046$, $P_{(Sqstm1)} = 0.0015$, $P_{(Optn)} = 0.0115$, $P_{(Tank)} = 0.0041$, $P_{(Scimp)} = 0.0056$, $P_{(Sarm1)} = 0.0029$, $P_{(Pde4d)} = 0.0026$, $P_{(Cd40)} = 1.9E-05$, $P_{(Tnf)} = 0.0022$, $P_{(Il-1b)} = 0.0056$, $P_{(Cxcl9)} = 0.0132$, $P_{(Cxcl10)} = 0.0064$. **c** Immunostaining of OPTN (red) and CD11c (green) in LPS treated or control human MoDCs and western blotting for OPTN in BMDCs stimulated with LPS or poly I:C for 8 h. Arrows indicate colabeled cells. Scale bar, 40 μm. $n = 3$ independent experiments. **d** qRT-PCR and western blotting to verify the deficiency of OPTN in DCs. $n = 3$ independent experiments, $P = 1.0E-11$. **e** Flow cytometry analysis of the frequency of total DCs (CD11c$^+$) in the bone marrow and spleen of Ctrl and $Optn^{\Delta CD11c}$ mice. $n = 3$ independent animals, $P_{(Spleen)} = 0.3289$, $P_{(Marrow)} = 0.5035$. **f, g** BMDCs from Ctrl and $Optn^{\Delta CD11c}$ mice were stimulated without (**f**) or with LPS (**g**). Expression levels of CD80, CD86, MHC-I, and MHC-II were analyzed by flow cytometry. $n = 3$ independent experiments, $P_{(CD80)} = 0.2275$, $P_{(CD86)} = 0.1680$ in (**f**); $P_{(CD80)} = 0.0034$, $P_{(CD86)} = 0.0015$, $P_{(MHC-I)} = 0.0017$, $P_{(MHC-II)} = 6.9E-05$ in (**g**). Data are presented as means ± SD. *$P < 0.05$; **$P < 0.01$; ***$P < 0.001$; NS, not significant. Unpaired two-tailed Student's $t$-test. Source data are provided in Source data file.

**Mice lacking *Optn* in CD11c$^+$ cells are resistant to auto-immunity.** We next employed a classical central nervous system (CNS) autoimmunity animal model EAE to explore the role of OPTN in regulating inflammatory injuries in vivo. Western blotting results showed that BMDCs from EAE mice expressed a higher level of OPTN than those from control mice (Fig. 3a). $Optn^{\Delta CD11c}$ mice displayed substantially decreased sensitivity to EAE induction and milder symptom compared with control mice, as indicated by lower clinical scores (Fig. 3b). As expected, $Optn^{\Delta CD11c}$ mice exhibited a less demyelinated

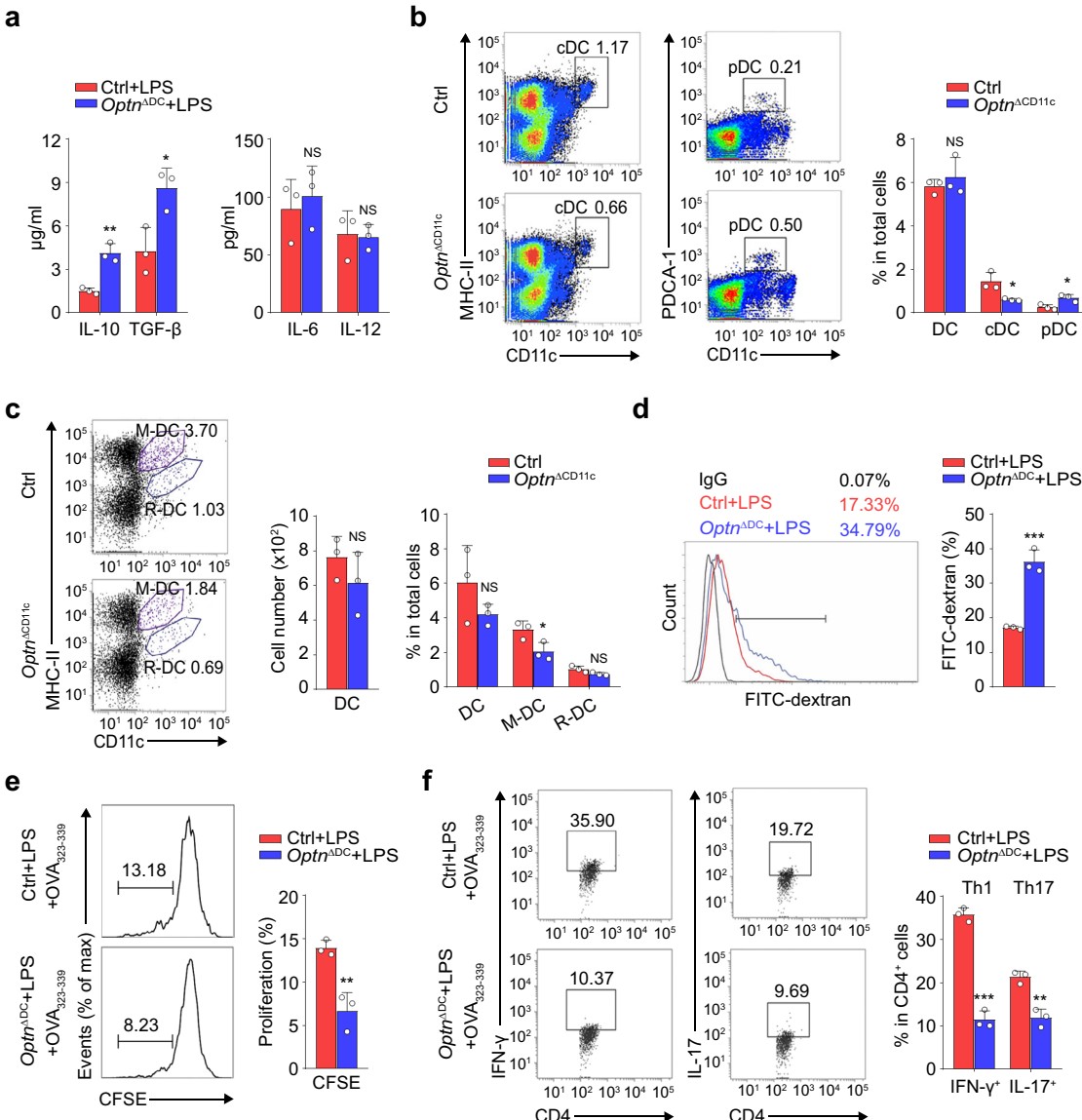

**Fig. 2 OPTN regulates the T cells priming function of DC. a** ELISA of cytokines in supernatants of LPS treated Ctrl and *Optn* null BMDCs. $n = 3$ independent experiments, $P_{(IL-10)} = 0.0029$, $P_{(TGF-\beta)} = 0.0239$, $P_{(IL-6)} = 0.6114$, $P_{(IL-12)} = 0.8559$. **b** Flow cytometry analysis of the frequency of total DC (CD11c$^+$), conventional DC (cDC, CD11c$^+$ MHCII$^+$) and plasmacytoid DCs (pDC, CD11c$^+$ PDCA-1$^+$) in the spleen of Ctrl and *Optn*$^{\Delta CD11c}$ mice. $n = 3$ independent animals, $P_{(DC)} = 0.4925$, $P_{(cDC)} = 0.0271$, $P_{(pDC)} = 0.0216$. **c** Flow cytometry analysis of the frequency of total DC (CD11c$^+$), migratory (M-DC, MHCII$^{hi}$ CD11c$^+$) and resident (R-DC, MHCII$^{int}$ CD11c$^+$) DCs in the draining lymph nodes of Ctrl and *Optn*$^{\Delta CD11c}$ mice. $n = 3$ independent animals, $P_{(DC number)} = 0.2950$, $P_{(DC\%)} = 0.2297$, $P_{(M-DC)} = 0.0394$, $P_{(R-DC)} = 0.0776$. **d** Uptake of FITC-dextran was measured by flow cytometry in Ctrl and *Optn* null BMDCs. $n = 3$ independent experiments, $P = 0.0007$. **e** Proliferation rate of CFSE-labeled OT-II CD4$^+$ T cells incubated with Ctrl or *Optn* null BMDCs pulsed with OVA$_{323-339}$. $n = 3$ independent experiments, $P = 0.0055$. **f** The percentages of differentiated Th1 and Th17 subsets of OT-II CD4$^+$ T cells incubated with Ctrl or *Optn* null BMDCs pulsed with OVA$_{323-339}$. $n = 3$ independent experiments, $P_{(IFN-\gamma)} = 8.6E-05$, $P_{(IL-17)} = 0.0028$. Data are presented as means ± SD. *$P < 0.05$; **$P < 0.01$; ***$P < 0.001$; NS, not significant. Unpaired two-tailed Student's *t*-test. Source data are provided in Source data file.

area and an increased number of CC1$^+$ mature oligodendrocytes in the white matter of spinal lesions at the peak of EAE (Fig. 3c, d), which could ultimately promote remyelination. In addition, *Optn*$^{\Delta CD11c}$ mice displayed a reduced frequency of inflammatory Th1 (CD4$^+$IFN-γ$^+$) and Th17 (CD4$^+$IL-17$^+$) cells as well as an increased amount of Th2 (CD4$^+$IL-4$^+$) and Treg (CD4$^+$Foxp3$^+$) cells in both spleen and DLNs (Fig. 3e, f). Likewise, the absolute numbers of mononuclear cells (MNCs) infiltrating into the CNS were lower in *Optn*$^{\Delta CD11c}$ mice than that in control mice (Fig. 3g). Taken together, these data indicate that depletion of *Optn* in CD11c$^+$ cells protects mice from CNS autoimmunity.

*Optn*$^{\Delta CD11c}$ **mice may have normal functions of T cells and macrophages**. Although DNA recombination in CD11c-cre line is mainly present in DCs, previous reports have shown that *CD11c*-Cre line has germline recombination and off-target problems, which may cause partial knockout of target genes in T cells and macrophages[18,19]. Consistently, we also found a weak deletion of OPTN in CD4$^+$ T cells but similar expression of OPTN in bone marrow-derived macrophages (BMDMs) from *Optn*$^{\Delta CD11c}$ mice (Supplementary Fig. 2a). Considering the significant role of DCs in T-cell activation, we then want to define the intrinsic impact of T cells in *Optn*$^{\Delta CD11c}$ mice. *Optn* knockout by *CD11c*-Cre did not influence T cells in the spleen and DLN, as the

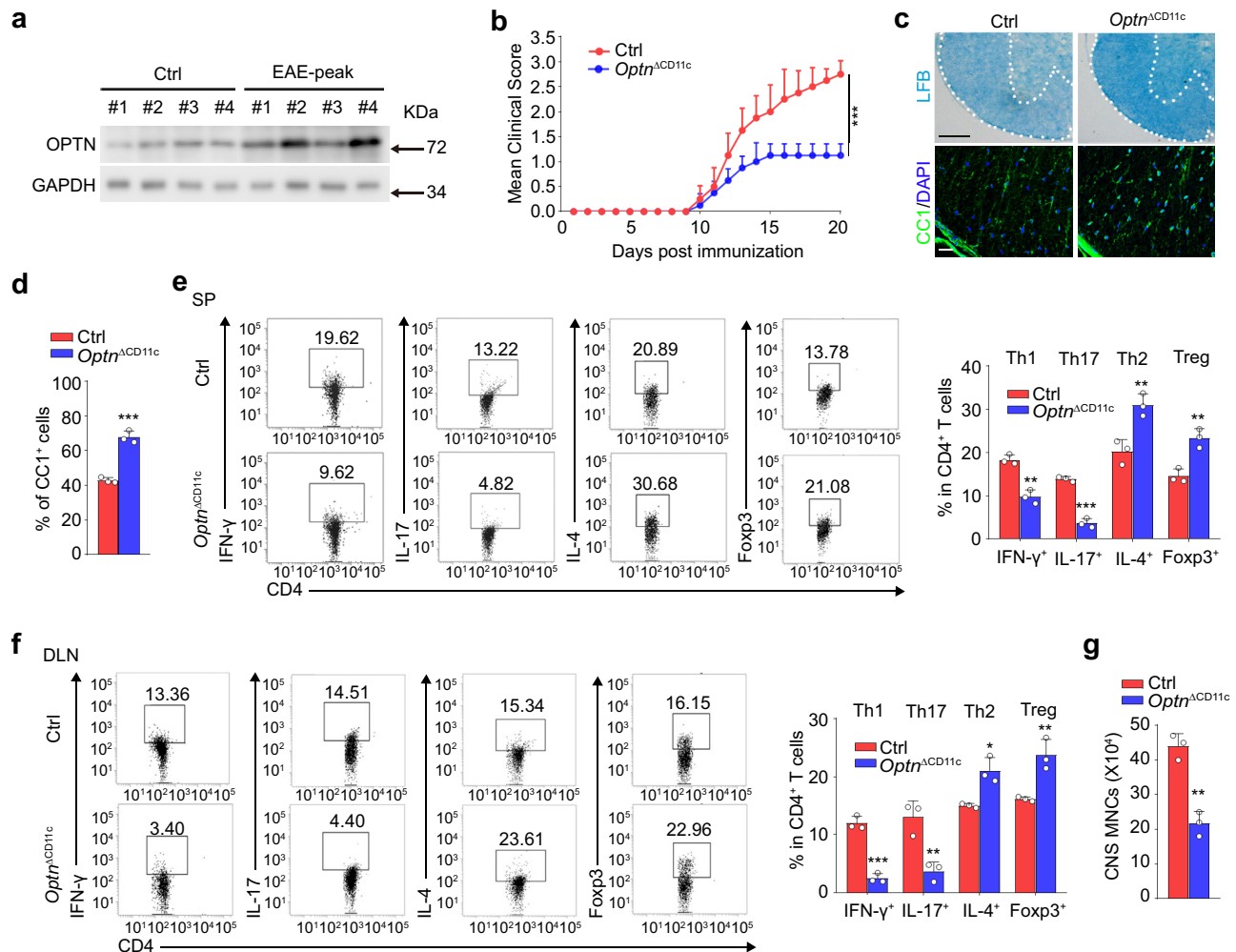

**Fig. 3 *Optn* deficient mice are resistant to EAE pathogenesis and inflammation. a** Western blotting analysis for the expression of OPTN in the BMDCs from four pairs of C57BL/6 Ctrl and EAE mice (8-week-old, female) at day 20 after immunization. $n = 3$ independent experiments. **b** Mean clinical scores of Ctrl and $Optn^{\Delta CD11c}$ mice (8-week-old, female) subjected to $MOG_{35-55}$-induced EAE. $n = 8$ independent animals, $P = 3.3E-09$. **c** (Upper) Luxolfast blue (LFB) staining of mouse spinal cords from Ctrl and $Optn^{\Delta CD11c}$ EAE mice at day 20 post immunization (dpi 20). Scale bars, 200 μm. (Lower) Immunofluorescent labeling of CC1 (green) and DAPI (blue) in the spinal cord of indicated mice at day 20 after immunization. Scale bars, 20 μm. $n = 3$ independent animals. **d** The summary graphs of $CC1^+$ cells in (**c**). $n = 3$ independent animals, $P = 0.0004$. **e, f** Frequency of Th1, Th17, Th2, and Treg cells in the spleen (SP) (**e**) and draining lymph nodes (DLN) (**f**) of $MOG_{35-55}$-immunized Ctrl and $Optn^{\Delta CD11c}$ mice on dpi 20. $n = 3$ independent animals, $P_{(IFN-\gamma)} = 0.0016$, $P_{(IL-17)} = 0.0001$, $P_{(IL-4)} = 0.0082$, $P_{(Foxp3)} = 0.0051$ in (**e**); $P_{(IFN-\gamma)} = 0.0003$, $P_{(IL-17)} = 0.0075$, $P_{(IL-4)} = 0.0103$, $P_{(Foxp3)} = 0.0088$ in (**f**). **g** The total numbers of MNCs in whole spinal cord and brain were isolated from Ctrl and $Optn^{\Delta CD11c}$ EAE mice on dpi 20. $n = 3$ independent animals, $P = 0.0015$. Data are presented as means ± SD. *$P < 0.05$; **$P < 0.01$; ***$P < 0.001$. Unpaired two-tailed Student's *t*-test. Source data are provided in Source data file.

comparative numbers of $CD4^+$ and $CD8^+$ T cells in $Optn^{\Delta CD11c}$ and control mice (Supplementary Fig. 2b). In addition, no infiltration of lymphocytes in the liver, lung, and spleen from $Optn^{\Delta CD11c}$ mice was observed (Supplementary Fig. 2c), coupled with the similar frequency of activated ($CD44^{hi}CD62L^{lo}$) $CD4^+$ and $CD8^+$ T cells in the spleen (Supplementary Fig. 2d, e). These results demonstrate that *Optn* knockout by *CD11c*-Cre does not influence T cells homeostasis in the immune system.

Given that EAE is mainly driven by $CD4^+$ T-cell activation[20], to rule out the role of *Optn* in $CD4^+$ T cells from $Optn^{\Delta CD11c}$ mice, we isolated $CD4^+$ T cells from WT and $Optn^{\Delta CD11c}$ mice, co-incubated with DCs from WT mice pulsed with $MOG_{35-55}$ for 2 days. FACS analysis showed that the percentage of Th1, Th2, Th17, and Treg cells were comparable between WT and $Optn^{\Delta CD11c}$ mice-derived $CD4^+$ T cells (Supplementary Fig. 2f), indicating that slight deletion of *Optn* had no effect on $CD4^+$ T-cell activation. To further clarify the role of $Optn^{\Delta CD11c}$ mice-

derived $CD4^+$ T cells in the in vivo phenotype of EAE mice, we then adoptively transferred $CD4^+$ T cells that were treated by WT DCs pulsed with $MOG_{35-55}$ to induce EAE. As expected, results indicated that WT and $Optn^{\Delta CD11c}$ mice-derived $CD4^+$ T cells had similar abilities to induce EAE pathogenesis (Supplementary Fig. 2g–i). Collectively, although we cannot completely rule out the role of OPTN in T cells, our results suggest that $CD4^+$ T cells derived from $Optn^{\Delta CD11c}$ mice do not affect EAE progression.

On the other hand, BMDMs from WT and $Optn^{\Delta CD11c}$ mice had comparable capacities on M1/M2 polarization and pro/anti-inflammatory cytokine expression (Supplementary Fig. 3a-g). Inspired by the findings that BMDMs can also express CD11c[21], our flow cytometry results showed that BMDMs from $Optn^{\Delta CD11c}$ mice had similar expression of CD11c to that of WT mice (Supplementary Fig. 3h). Besides, the polarization ability of $F4/80^+CD11c^+$ BMDMs from WT and $Optn^{\Delta CD11c}$ mice was also comparable (Supplementary Fig. 3i). Thus,

macrophages from $Optn^{\Delta CD11c}$ mice may have similar function to that from WT mice in vitro. Even so, there are still drawbacks that the potential function of OPTN in CD11c+ macrophages in vivo is still unknown and whether CD11c+ macrophages in $Optn^{\Delta CD11c}$ mice influent EAE pathology is also undefined. Nonetheless, our data at least clarify the function of OPTN in DCs and prove that $Optn$ deficiency in CD11c+ cells mitigates EAE effectively.

**STAT3 signaling pathway participates in $Optn$ deficient DC maturation**. STAT3, among all the members of the STAT family, has been well documented to promote the abnormal DC differentiation and function[22]. Consistently, we also found that phosphorylated STAT3 (p-STAT3), a critical transcription factor for STAT3 transduction, was elevated after LPS treatment for 16 h (Fig. 4a). However, a subsequent western blotting analysis showed that $Optn$ deficiency activated STAT3 transduction prominently after LPS treatment for just 8 h (Fig. 4b), which was further confirmed by the increased p-STAT3 level in nuclear subfractions of $Optn$ null BMDCs (Fig. 4c). Although STAT3 in macrophages was found to limit the inflammation of EAE mice[23], our results showed that BMDMs from $Optn^{\Delta CD11c}$ mice had similar expression of (p-)STAT3 to WT mice, and the level of p-STAT3 was elevated fairly on IL-4 stimulation, further indicating that $Optn^{\Delta CD11c}$ may have similar macrophage function (Supplementary Fig. 3j). Generally, STAT3 can transcriptionally regulate the expression of many classical target genes like $Vegfa$, $Hif1a$, $Hgf$, $Ptgs2$, and $Socs3$, as well as many cytokine genes like $Il-10$, $Il-6$, $Il-13$, and $Tgfb$[24–27]. And many of them, such as VEGF, HIF-1α, HGF, SOCS3, IL-10, and IL-13, have been verified in previous studies to inhibit LPS-dependent DC maturation or induce the differentiation of DCs into tolerogenic DCs[28–32]. Interestingly, we found that most of these target genes were significantly enriched in $Optn$ knockout BMDCs (Fig. 4d), suggesting that STAT3 signaling pathway was activated in an early stage in $Optn$ deficient DC maturation.

We next explored if STAT3 participates in OPTN-mediated DC maturation and activation. Notably, STAT3 overexpression significantly reduced the mRNA levels of DC co-stimulatory molecules and MHC molecules (Fig. 4e, f), consistent with previous findings that STAT3 is crucial for the tolerant function of DCs and DCs conditional deletion of $Stat3$ resulted in T-cell activation and autoimmunity in mice[33]. On the contrary, a pharmacological inhibitor of STAT3 phosphorylation, stattic, treatment reduced the p-STAT3 level in both control and $Optn$-deficient DCs (Fig. 4g). qRT-PCR and flow cytometry analysis indicated that stattic treatment not only promoted mature DCs markers CD80 and MHC-II expression in control BMDCs, but also rescued the impaired maturation and activation in $Optn$ knockout BMDCs (Fig. 4h–j). These data suggest that STAT3 has an important role in $Optn$ deficient DC maturation.

The well-characterized functions of OPTN are autophagy and ubiquitin binding[11], so we then checked if these functions may be related to OPTN-regulated DC maturation. First, although $Optn$ deletion inhibited the rapamycin (RAPA) activated GFP+mCherry+ autophagic puncta (Supplementary Fig. 4a, b), $Optn$ knockout cannot rescue RAPA mediated depressed expression of mature DCs markers CD80 and CD86 (Supplementary Fig. 4c, d). On the other hand, we constructed a whole OPTN plasmid and an OPTN plasmid with UBD deletion (OPTN$^{\Delta UBD}$), and transfected them into $Optn$ null BMDCs (Supplementary Fig. 4e). Flow cytometry results showed that both OPTN and OPTN$^{\Delta UBD}$ overexpression restored the percentage of CD11c+CD80+ and CD11c+CD86+ cells in LPS-stimulated $Optn$ null BMDCs (Supplementary Fig. 4f, g). Moreover, OPTN and OPTN$^{\Delta UBD}$

had similar ability to reverse the mRNA level of $Il-10$ in $Optn$ deficient BMDCs (Supplementary Fig. 4h). In conclusion, these results suggest that OPTN-mediated autophagy or ubiquitination has no contribution to OPTN-regulated DC maturation.

**STAT3 is crucial for the anti-inflammatory effects of $Optn^{\Delta CD11c}$ mice**. Given the essential role of STAT3 in DC maturation, we generated CD11c+ cells conditional $Stat3$ knockout mice ($Stat3^{fl/fl}$; $CD11c$-Cre mice, hereafter called $Stat3^{\Delta CD11c}$ mice) to investigate if STAT3 in CD11c+ cells limits EAE. Adoptive transfer study showed that $Stat3$ deficient BMDCs aggravated MOG$_{35-55}$ induced EAE progression when compared with control BMDCs (Supplementary Fig. 5). To further determine whether the immune tolerance of $Optn^{\Delta CD11c}$ mice is dependent on STAT3 signaling, we crossed the $Optn^{\Delta CD11c}$ mice with $Stat3^{fl/fl}$ mice ($Optn^{fl/fl}$; $Stat3^{fl/fl}$; $CD11c$-Cre mice, hereafter called DKO mice) and verified the knockout efficiency by western blotting (Fig. 5a). Results showed that DKO mice displayed much more severe autoimmune symptoms than $Optn^{\Delta CD11c}$ mice, as evidenced by higher clinical score, more severe demyelination, and less CC1+ mature oligodendrocytes (Fig. 5b–d). In addition, the percentages of Th1 and Th17 cells were much higher in both the spleen and DLN of DKO mice during EAE than those of $Optn^{\Delta CD11c}$ mice (Fig. 5e, f). Conversely, the amounts of Th2 and Treg cells in DKO mice were significantly decreased when compared with $Optn^{\Delta CD11c}$ mice (Fig. 5e, f). Together, these results suggest that STAT3 in CD11c+ cells is a negative regulator of autoimmune response, and OPTN can manipulate STAT3 activity in CD11c+ cells to control autoimmune progress.

**OPTN restrains the phosphorylated activation of JAK2 in DCs**. Encouraged by the findings, we proceeded to study how OPTN regulates STAT3 signaling pathway. We found JAK2, the classic upstream signal of STAT3, was also activated observably after LPS treatment for only 8 h (Fig. 6a). To find out whether OPTN modulates JAK2 phosphorylated activation, we overexpressed OPTN in control and $Optn$ deficient BMDCs. Results showed that OPTN abundance could inhibit the expression of p-JAK2 in both control and $Optn$ knockout BMDCs stimulated with LPS or poly I:C (Fig. 6b, c). Considering that IL-6 and IL-10, classical stimulators of JAK2, are rapidly released in response to TLR4/NF-κB signals stimulated by LPS[34,35], we therefore suspected that IL-6/IL-10 may be involved in the regulation of JAK2-STAT3 in DCs. Despite the fact that both anti-IL6 and anti-IL10 eliminated LPS induced JAK2-STAT3 activation in $Optn$-deficient DCs (Fig. 6d), there was no direct interaction between TLR4 and JAK2 or STAT3 in DCs after LPS treatment and no marked difference in NF-κB signals between control and $Optn$-deficient DCs (Fig. 6e, f). In consequence, these data indicate that LPS activated JAK2/STAT3 is dependent on the IL-6/IL-10 but not the recruitment of JAK2 or STAT3 to TLR4, and OPTN deletion cannot interfere the NF-κB signals which manipulate the initial transcription of IL-6/IL-10 in LPS treatment.

We then asked whether $Optn$ knockdown could cause the constitutive activation of JAK2 and employed IL-6 to activate the IL-6 receptor/JAK2 signaling pathway. As a result, OPTN overexpression decreased the phosphorylation level of JAK2, while $Optn$ depletion augmented the JAK2 activation (Fig. 6g). More importantly, this effect was strengthened after IL-6 treatment (Fig. 6g). Finally, the treatment of RAPA did not alter the level of p-JAK2 and p-STAT3 (Fig. 6h), proving that the activation of JAK2/STAT3 in $Optn$ knockout BMDCs was independent of the OPTN-mediated autophagy deficiency.

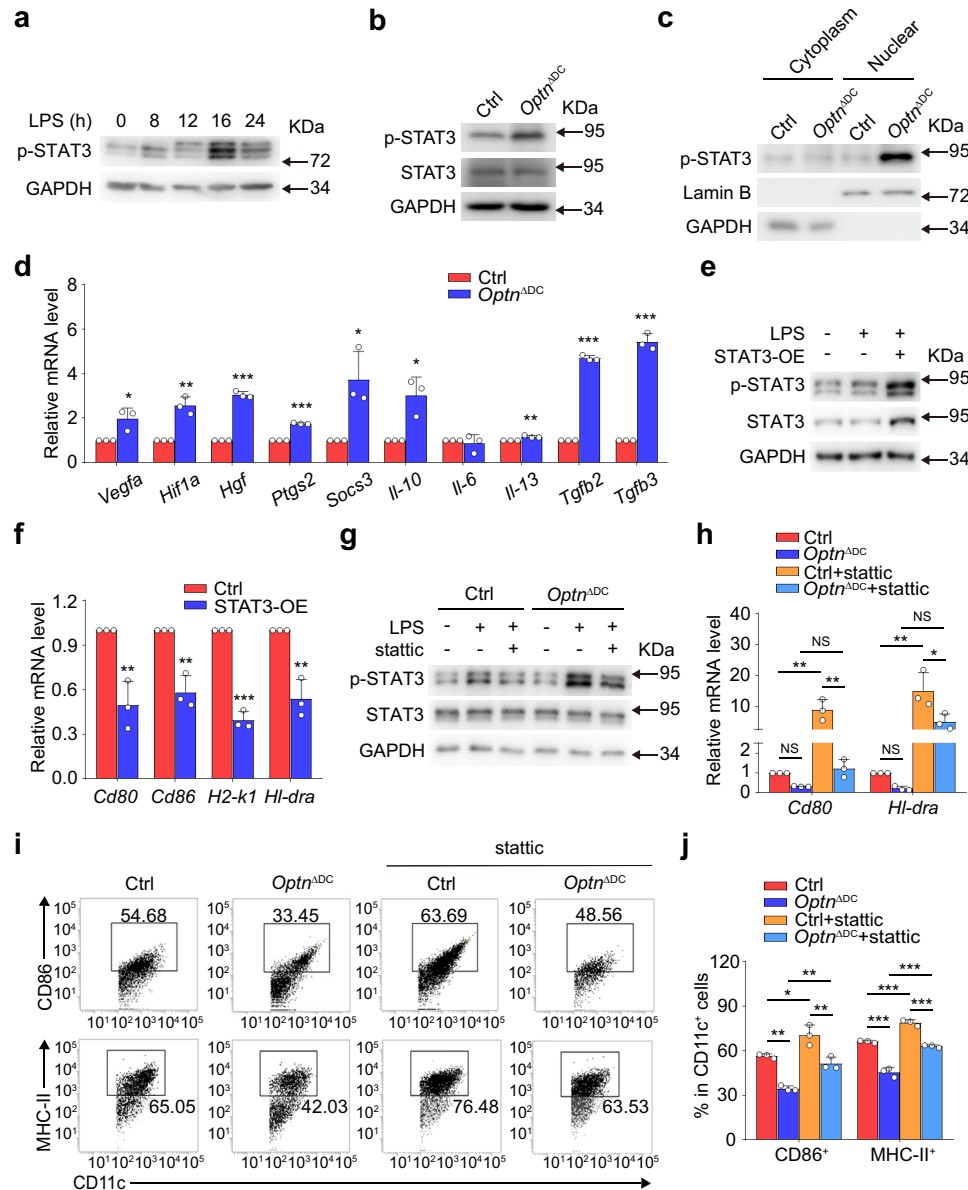

**Fig. 4 STAT3 participates in OPTN-regulated DC maturation. a** Expression of p-STAT3 in BMDCs stimulated with LPS for different time. $n = 3$ independent experiments. **b** Expression of (p)-STAT3 in Ctrl or *Optn* knockout BMDCs stimulated with LPS for 8 h. $n = 3$ independent experiments. **c** Cytoplasmic and nuclear proteins from Ctrl and *Optn* knockout BMDCs were analyzed by western blotting. The nuclear marker Lamin B and the cytoplasmic marker GAPDH were used to demonstrate the purity of the fractions. $n = 3$ independent experiments. **d** qRT-PCR analysis of STAT3 target genes in Ctrl and *Optn* knockout BMDCs. $n = 3$ independent experiments, $P_{(Vegfa)} = 0.0238$, $P_{(Hif1a)} = 0.0023$, $P_{(Hgf)} = 1.8E-05$, $P_{(Ptgs2)} = 2.1E-06$, $P_{(Socs3)} = 0.0205$, $P_{(Il-10)} = 0.0137$, $P_{(Il-13)} = 0.0076$, $P_{(Tgfb2)} = 8.9E-07$, $P_{(Tgfb3)} = 3.1E-05$. **e**, **f** Ctrl BMDCs were transfected with STAT3-overexpression (OE) plasmid. mRNA level of *Cd80*, *Cd86*, *H2-k1*, and *Hl-dra* were analyzed by qRT-PCR. $n = 3$ independent experiments, $P_{(Cd80)} = 0.0056$, $P_{(Cd86)} = 0.0031$, $P_{(H2-k1)} = 6.4E-05$, $P_{(Hl-dra)} = 0.0034$. **g-j** Ctrl and *Optn* knockout BMDCs were treated with stattic (10 μM) before LPS stimulation. Expression of *Cd80*, *Hl-dra* (**h**) and CD86, MHC-II (**i**, **j**) were analyzed by qRT-PCR and flow cytometry, respectively. $n = 3$ independent experiments, $P_{(Optn^{\Delta DC}-Ctrl)} = 0.9612$, $P_{(Ctrl+stattic-Ctrl)} = 0.0025$, $P_{(Optn^{\Delta DC}+stattic-Optn^{\Delta DC})} = 0.9157$, $P_{(Optn^{\Delta DC}+stattic-Ctrl+stattic)} = 0.0030$ of *Cd80* in (**h**); $P_{(Optn^{\Delta DC}-Ctrl)} = 0.9902$, $P_{(Ctrl+stattic-Ctrl)} = 0.0031$, $P_{(Optn^{\Delta DC}+stattic-Optn^{\Delta DC})} = 0.3368$, $P_{(Optn^{\Delta DC}+stattic-Ctrl+stattic)} = 0.0208$ of *Hl-dra* in (**h**); $P_{(Optn^{\Delta DC}-Ctrl)} = 0.00105$, $P_{(Ctrl+stattic-Ctrl)} = 0.0153$, $P_{(Optn^{\Delta DC}+stattic-Optn^{\Delta DC})} = 0.0057$, $P_{(Optn^{\Delta DC}+stattic-Ctrl+stattic)} = 0.0024$ of CD86 in (**j**); $P_{(Optn^{\Delta DC}-Ctrl)} = 6.5E-06$, $P_{(Ctrl+stattic-Ctrl)} = 0.0002$, $P_{(Optn^{\Delta DC}+stattic-Optn^{\Delta DC})} = 2.2E-05$, $P_{(Optn^{\Delta DC}+stattic-Ctrl+stattic)} = 4.7E-05$ of MHC-II in (**j**). Data are presented as means ± SD. \*$P < 0.05$; \*\*$P < 0.01$; \*\*\*$P < 0.001$; NS, not significant. Unpaired two-tailed Student's *t*-test for (**d**, **f**); one-way ANOVA Tukey's post hoc analysis for (**h**, **j**). Source data are provided in Source data file.

Overall, our results confirm that OPTN negatively modulates JAK2 phosphorylated activation in DCs.

**OPTN inhibits JAK2 dimerization and subsequent STAT3 activation.** Next, we wonder how OPTN regulates JAK2/STAT3 signaling. Immunoprecipitation analysis showed that ectopically expressed OPTN was reciprocally precipitated with JAK2, but not JAK1 or JAK3, from HEK293 cells overexpressing the proteins (Fig. 7a). A subsequent co-immunoprecipitation further confirmed the direct interaction between endogenous OPTN and JAK2 in mouse primary BMDCs (Fig. 7b). JAK2 kinase is composed of 7 JAK homology (JH) domains, termed JH1-7. The JH1 region functions as the kinase domain of JAK2, while JH2 can physically interact with JH1 and inhibit its kinase

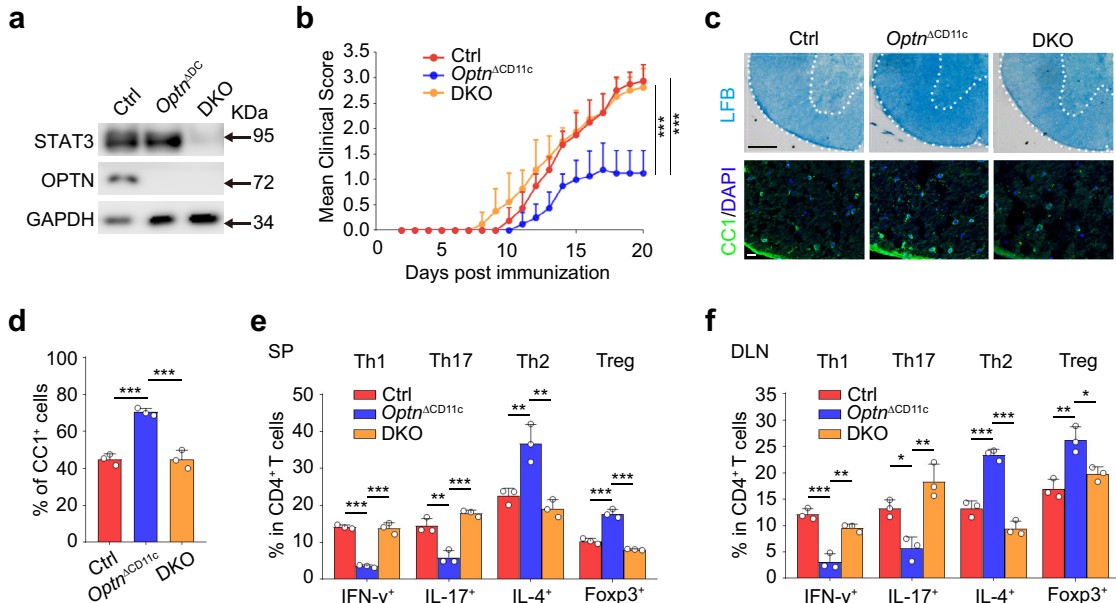

**Fig. 5 Deletion of *Stat3* aggravates autoimmunity in *Optn*$^{\Delta CD11c}$ mice. a** Western blotting to verify the deficiency of OPTN and STAT3 in DKO DCs. $n = 3$ independent experiments. **b** Mean clinical scores of Ctrl, *Optn*$^{\Delta CD11c}$ and DKO mice (8-week-old, female) subjected to MOG$_{35-55}$-induced EAE. $n = 8$ independent animals, $P_{(Ctrl)} = 5.1E{-}08$, $P_{(DKO)} = 1.6E{-}07$. **c** (Upper) Luxolfast blue (LFB) staining of spinal cords from indicated EAE mice at day 20. Scale bars, 200 μm. (Lower) Immunofluorescent labeling of CC1 (green) and DAPI (blue) in the spinal cord of indicated mice at day 20. Scale bars, 20 μm. $n = 3$ independent animals. **d** The summary graphs of CC1$^+$ cells in (**c**). $n = 3$ independent animals, $P_{(Ctrl)} = 0.0003$, $P_{(DKO)} = 0.0003$. **e, f** Frequency of Th1, Th17, Th2, and Treg cells in the spleen (SP) (**e**) and draining lymph nodes (DLN) (**f**) from indicated EAE mice on day 20 after immunization. $n = 3$ independent animals, $P_{(Ctrl)} = 2.3E{-}05$, $P_{(DKO)} = 2.8E{-}05$ of IFN-γ in (**e**); $P_{(Ctrl)} = 0.0018$, $P_{(DKO)} = 0.0003$ of IL-17 in (**e**); $P_{(Ctrl)} = 0.0059$, $P_{(DKO)} = 0.0020$ of IL-4 in (**e**); $P_{(Ctrl)} = 7.1E{-}05$, $P_{(DKO)} = 1.5E{-}05$ of Foxp3 in (**e**); $P_{(Ctrl)} = 0.0002$, $P_{(DKO)} = 0.00102$ of IFN-γ in (**f**); $P_{(Ctrl)} = 0.0215$, $P_{(DKO)} = 0.0018$ of IL-17 in (**f**); $P_{(Ctrl)} = 0.0002$, $P_{(DKO)} = 3.2E{-}05$ of IL-4 in (**f**); $P_{(Ctrl)} = 0.0026$, $P_{(DKO)} = 0.0155$ of Foxp3 in (**f**). Data are presented as means ± SD. *$P < 0.05$; **$P < 0.01$; ***$P < 0.001$. One-way ANOVA Tukey's post hoc analysis. Source data are provided in Source data file.

activity. And the JH3-7 region of JAK2 is essential for receptor interactions[36]. Accordingly, we generated a series of truncated JAK2 mutants to represent these domains. Analysis of the interaction between OPTN and the JAK2 mutants identified JH1, rather than JH2 or JH3-7, as the site for OPTN binding (Fig. 7c). Besides, the interaction between OPTN and JH1 did not rely on its UBD, as OPTN$^{\Delta UBD}$ can also interact with JAK2 normally (Fig. 7d). Therefore, these results suggest that OPTN may regulate JAK2 directly.

Previous studies suggested that JH1-JH1 trans-interaction mediates the dimerization and activation of JAK2[37], which prompted us to investigate whether OPTN inhibits JAK2 dimerization. The results showed that the ability of FLAG-tagged JAK2 to co-immunoprecipitate with HA-tagged JAK2 was markedly hampered by the overexpression of OPTN (Fig. 7e). Moreover, the disuccinimidyl suberate-mediated cross-linking assays indicated that the dimer/monomer ratio of JAK2 was decreased in the presence of OPTN in both HEK293 and BMDCs (Fig. 7f, g), further showing the inhibitory ability of OPTN to JAK2 dimerization. Given that once activated, JAK2 recruits and phosphorylates STAT3, we then wondered whether OPTN could affect JAK2 downstream activation. As expected, the JAK2-STAT3 interaction was abrogated by OPTN overexpression in both HEK293 and BMDCs (Fig. 7h, i), implying that OPTN antagonized the binding of JAK2 with STAT3, further interfering STAT3 activation.

STAT3, a classical transcription factor, can transfer to the nucleus and bind to its target genes like *Il-10* after phosphorylated activation[27]. To further explore the transcriptional regulatory role of STAT3, we analyzed the public STAT3 ChIP-seq data from GEO dataset[38]. Peak tracking results revealed that STAT3 had quite high bindings with *Il-10* gene in DCs, while STAT3 showed

weak bindings with genes like *Il-6*, *Cd80*, *Cd86*, *Mhc-i* (*H2-K1*), *Mhc-ii* (*H2-Aa*), *Vegfa*, *Hif1a*, *Hgf*, *Ptgs2*, *Il-13*, *Tgfb2*, or *Tgfb3* (Fig. 7j and Supplementary Fig. 6), suggesting IL-10 might be the effective target gene of STAT3 in DCs. Consistent with the previous studies[35,39], we found that the expression of *Il-6* and *Il-10* was increased and then went down after LPS treatment in control BMDCs (Fig. 7k). Inspiringly, knocking out of *Optn* in BMDCs facilitated and maintained the expression of IL-10 but not IL-6 (Figs. 2a and 7k), thus exaggerating the immunosuppressive function of IL-10[31], and resulting in the evident activation of JAK2/STAT3 in *Optn* knockout BMDCs. Collectively, our data demonstrate that OPTN negatively regulates JAK2/STAT3 activation by interacting with JAK2 and inhibiting its dimerization, and *Optn* deficiency hinders DC maturation via activating IL-10/JAK2/STAT3/IL-10 positive feedback loop.

In addition to *Il-10* target, we also found STAT3 could bind to its canonical target *Socs3* gene in DCs by STAT3 ChIP-seq data peak tracking (Supplementary Fig. 6), and our qPCR results showed the gene expression of *Socs3* was significantly upregulated in *Optn* deficient BMDCs (Fig. 4d). Given that SOCS3 is found to inhibit DC maturation together with subsequent Th17 response[32], these data indicate that *Optn* deficiency can inhibit DC maturation at least partially via activating STAT3 target *Socs3*.

**SSD impairs DC function and alleviates EAE by reducing OPTN expression.** The promising effect of *Optn* deficiency on DCs dictated immune activation shed creative lights on finding treatments for autoimmune diseases. We screened the natural product library for small molecule inhibitors of OPTN expression and found that Saikosaponin D (SSD), a triterpenoid saponin derived from Bupleurum falcatum L, significantly downregulated

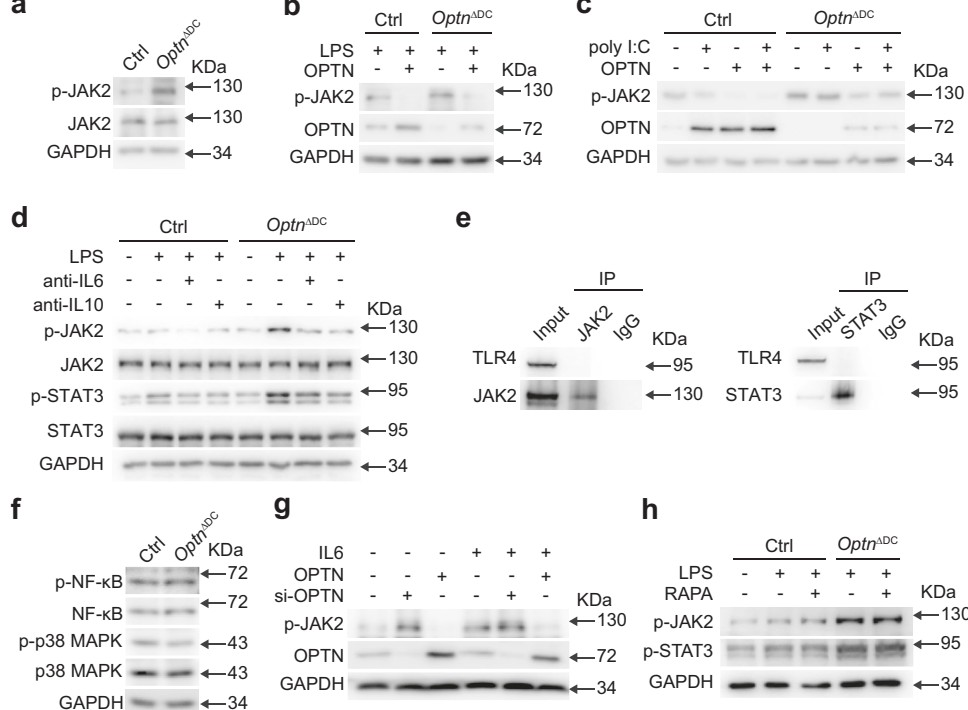

**Fig. 6 OPTN restrains the IL-6/IL-10 activated JAK2/STAT3 phosphorylation in DCs. a** Expression of (p)-JAK2 in Ctrl or *Optn* deficient BMDCs stimulated with LPS for 8 h. **b, c** Ctrl and *Optn* deficient BMDCs were stimulated with LPS (**b**) or poly I:C (**c**) after transfection with empty or OPTN expression vector. The expressions of OPTN and p-JAK2 were confirmed by western blotting. **d** Western blotting for expression of (p)-JAK2 and (p)-STAT3 in Ctrl and *Optn* deficient BMDCs stimulated with anti-IL6 or anti-IL10 for 48 h. **e** Interaction between endogenous TLR4 and JAK2 or STAT3 in BMDCs was analyzed by IP with anti-JAK2 or anti-STAT3 using normal rabbit IgG as a control, followed by immunoblotting with anti-TLR4, anti-JAK2, and anti-STAT3. **f** Expression of (p-) signaling proteins in whole-cell lysates of Ctrl or *Optn* deficient BMDCs stimulated with LPS. **g** HEK293 cells were transfected with *si-Optn* or *Optn* plasmid before IL-6 stimulation. The expressions of OPTN and p-JAK2 were confirmed by western blotting. **h** Western blotting for expression of p-JAK2 and p-STAT3 in Ctrl and *Optn* deficient BMDCs stimulated with RAPA. All data are representative of three independent experiments. Source data are provided in Source data file.

OPTN in BMDCs after LPS stimulation (Fig. 8a). To investigate how SSD inhibits OPTN expression, we next analyzed the binding profiles of SSD and OPTN by using surface plasmon resonance (SPR) assay. Results revealed that SSD specifically binds to OPTN (KD = $5.115 \times e^{-5}$ M), with rapid on-rate and off-rate (Supplementary Fig. 7a, b), but not ubiquitin-like modifier activating enzyme 3 (UBA3) (Supplementary Fig. 7c, d). Then we performed a cycloheximide (CHX) chase assay in SSD-treated BMDCs to determine whether SSD affects the stability of OPTN protein. Western blotting results indicated that the stability of OPTN was decreased in the presence of SSD (Fig. 8b), suggesting that the interaction of SSD with OPTN promotes the degradation of OPTN, thus reducing OPTN expression.

Next, the effect of SSD on DC activation was examined, and flow cytometry results showed that SSD markedly reduced the expression of CD80, CD86, and MHC-II in BMDCs (Fig. 8c, d). In addition, SSD-treated BMDCs exhibited an impaired ability to initiate the proliferation and differentiation of OT-II CD4+ T cells (Fig. 8e–g). These data indicate that SSD inhibits DC maturation and activation, which may be further linked to the potential therapeutic role of SSD on autoimmune disease. Our results showed that SSD treatment daily protected mice against the EAE challenge in a dose-dependent way (Fig. 8h). Luxolfast blue (LFB) and CC1 staining further confirmed that demyelination was markedly decreased and mature oligodendrocytes were dramatically increased after SSD treatment (Fig. 8i). Elsewise, the amount of the inflammatory Th1 and Th17 cells in peripheral lymphoid organs of SSD-treated mice were reduced significantly during disease development (Fig. 8j, k). All together, these results

suggest that SSD effectively inhibits not only the immune-stimulatory function of DCs but also the disease progression of EAE.

We then asked whether SSD specifically depends on OPTN to regulate DC maturation and EAE progression. Western blotting results showed that SSD had no effect on TLR4 signaling, as shown by the equivalent expression of TLR4, MyD88, (p-)NF-κB, (p-)IRF3, and (p-)p38 MAPK upon SSD treatment (Supplementary Fig. 7e). Meanwhile, qRT-PCR analysis revealed the comparable mRNA levels of *Cd80* and *Cd86* in *Optn* knockout BMDCs with or without SSD incubation (Supplementary Fig. 7f). Finally, SSD administration exhibited a similar therapeutic effect as *Optn* deletion in effectively inhibiting the progression of EAE, and had no effect on *Optn*$^{\Delta CD11c}$ mice undergoing EAE progress (Supplementary Fig. 7g). In aggregate, these results suggest that SSD ameliorates experimental autoimmunity through OPTN modulation.

## Discussion

Immunostimulatory properties of DCs are closely associated with their maturation state[2]. Tolerogenic immature DCs weaken T-cell functions and inhibit immune responses in an autoimmune disease, while mature DCs promote the immune activities in pathogenic conditions. However, the signaling network that mediates the maturation of DCs has not been fully understood. Meanwhile, emerging evidence highlight the pivotal role of OPTN in the immune system, but the mechanisms underlying the effects of OPTN on DC activation and immunogenicity

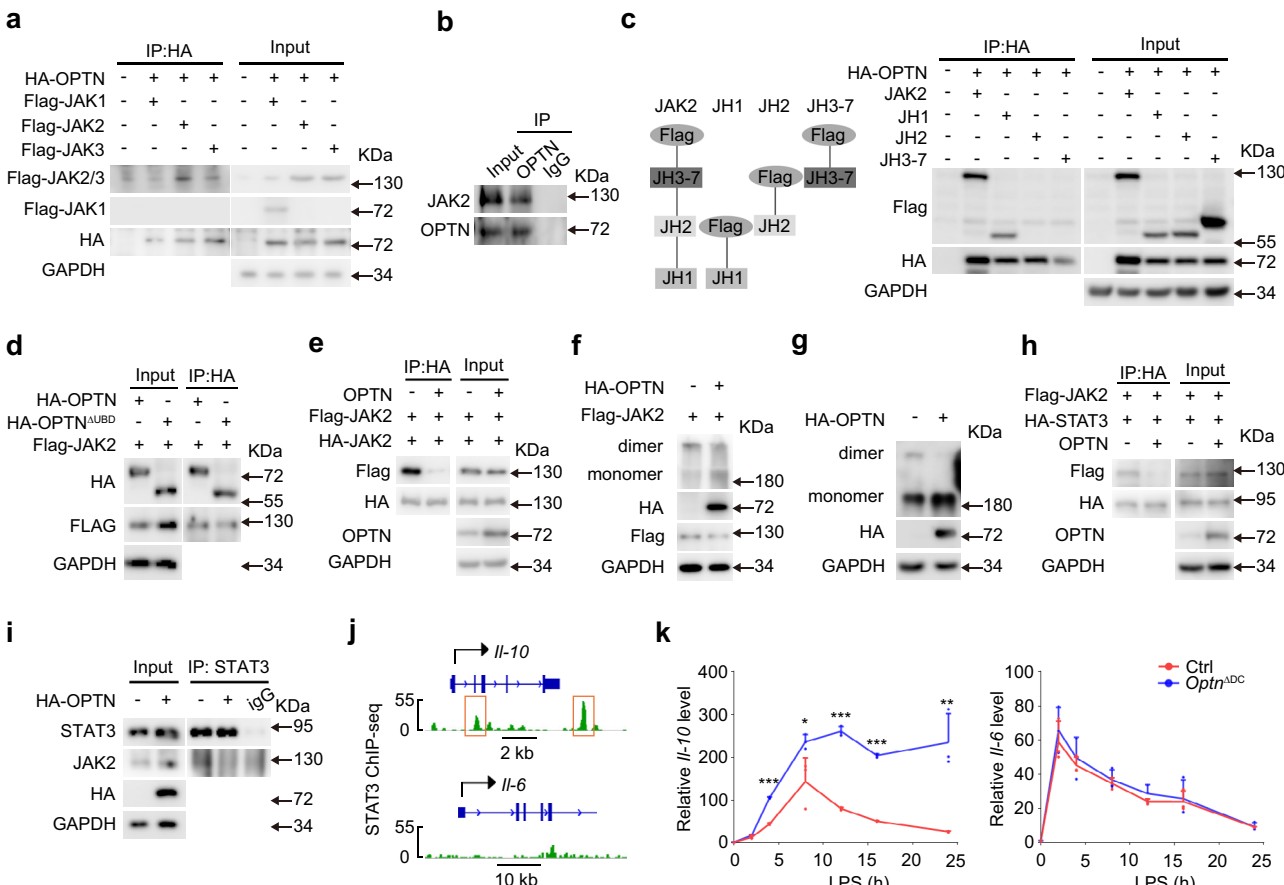

**Fig. 7 OPTN inhibits the dimerization of JAK2 and subsequent STAT3 activation. a** HA-OPTN expression plasmid was transfected with Flag-JAK1, Flag-JAK2, or Flag-JAK3 expression plasmid into HEK293 cells. Expression of OPTN and JAKs was confirmed by immunoblotting. Interaction between OPTN and JAKs was determined by immunoprecipitation (IP) with anti-HA antibody followed by immunoblotting with anti-Flag antibody. **b** Interaction between endogenous OPTN and JAK2 in BMDCs was analyzed by IP. **c** Schematic illustration of the truncated JAK2. Interactions between OPTN and truncation JAK2 in transiently transfected HEK293 cells were determined by IP. **d** Interactions between JAK2 with OPTN or OPTN$^{\Delta UBD}$ in transiently transfected HEK293 cells were determined by IP. **e** Interactions between HA-JAK2 and FLAG-JAK2 in transiently transfected HEK293 cells was determined by IP. **f, g** HEK293 cells were transfected with FLAG-JAK2 and HA-OPTN or empty vector (**f**). BMDCs were transfected with HA-OPTN or empty vector (**g**). Whole-cell lysates were incubated with disuccinimidyl suberate (DSS). JAK2 antibody marked two bands: the upper band referring to the dimer and the lower band representing the monomer of JAK2. **h** Interaction of FLAG-JAK2 and HA-STAT3 in transiently transfected HEK293 cells was determined by IP. **i** BMDCs were transfected with HA-OPTN or empty vector. Interaction between endogenous STAT3 and JAK2 in BMDCs was analyzed by IP. **j** ChIP-seq analysis of the binding between STAT3 and *Il-10* or *Il-6* from GSE27161. **k** *Il-10* and *Il-6* mRNA level in Ctrl and *Optn* deficient BMDCs stimulated with LPS for different times. $n = 3$ independent experiments, $P_{(4)} = 3.2E{-}06$, $P_{(8)} = 0.0497$, $P_{(12)} = 1.2E{-}05$, $P_{(16)} = 3.9E{-}07$, $P_{(24)} = 0.0060$ of *Il-10*. Data in (**a**–**i**) are representative of three independent experiments. Data are presented as means ± SD. *$P < 0.05$; **$P < 0.01$; ***$P < 0.001$. Unpaired two-tailed Student's *t*-test. Source data are provided in Source data file.

remains elusive. Our present study demonstrated that *Optn* deficiency hampered the antigen presenting function of DCs and their ability to activate CD4$^+$ T cells and differentiate into Th1 and Th17 subtypes, and *Optn* conditional knockout via *CD11c*-Cre in mice model could effectively disrupt the progression of EAE, a T cells dependent experimental autoimmune disease (Supplementary Fig. 8).

First of all, culture of DC precursors from bone marrow with GM-CSF or FLT3L are two common systems used in vitro. Of note, although both GM-CSF and FLT3L stimulated murine bone marrow cultures comprise a heterogeneous population of CD135$^+$ DCs and CD115$^+$ macrophages or monocyte progeny[15,40], our GM-CSF/IL-4 stimulated BMDC culture system contained almost no CD115$^+$ macrophages (Supplementary Fig. 1), which may be relevant to the use of high concentration of IL-4 (20 ng/ml)[15]. Importantly, it has been also reported that GM-CSF/IL-4 derived DCs are the equivalents of in vivo DCs that emerge after inflammation, whereas FLT3L

derived DCs better represent the steady-state resident DCs[41]. Taken together, GM-CSF/IL-4 is supposed to be a better one in our study for DC culture experiments.

What's more, although *CD11c*-Cre line is associated with germline recombination or off-target problem in T cells as the presence of CD11c on the minor subpopulation of them[18,19], our validation nonetheless showed that *Optn* deletion via *CD11c*-Cre did not affect the function of T cells. Besides, more and more studies have found that CD11c is also expressed in a portion of macrophages in many tissues like liver, adipose tissue, intestinal, and alveolar[42–44], the ratio of which will be significantly elevated in inflammatory conditions. Thus, CD11c$^+$ macrophages are claimed as activated, pro-inflammatory or M1-like monocytes derived macrophages[21]. Consistent with previous findings[21], we found that bone marrow monocytes derived BMDMs also contain CD11c$^+$ macrophages, the percentage and function of which was similar in BMDMs from Ctrl and *Optn*$^{\Delta CD11c}$ mice

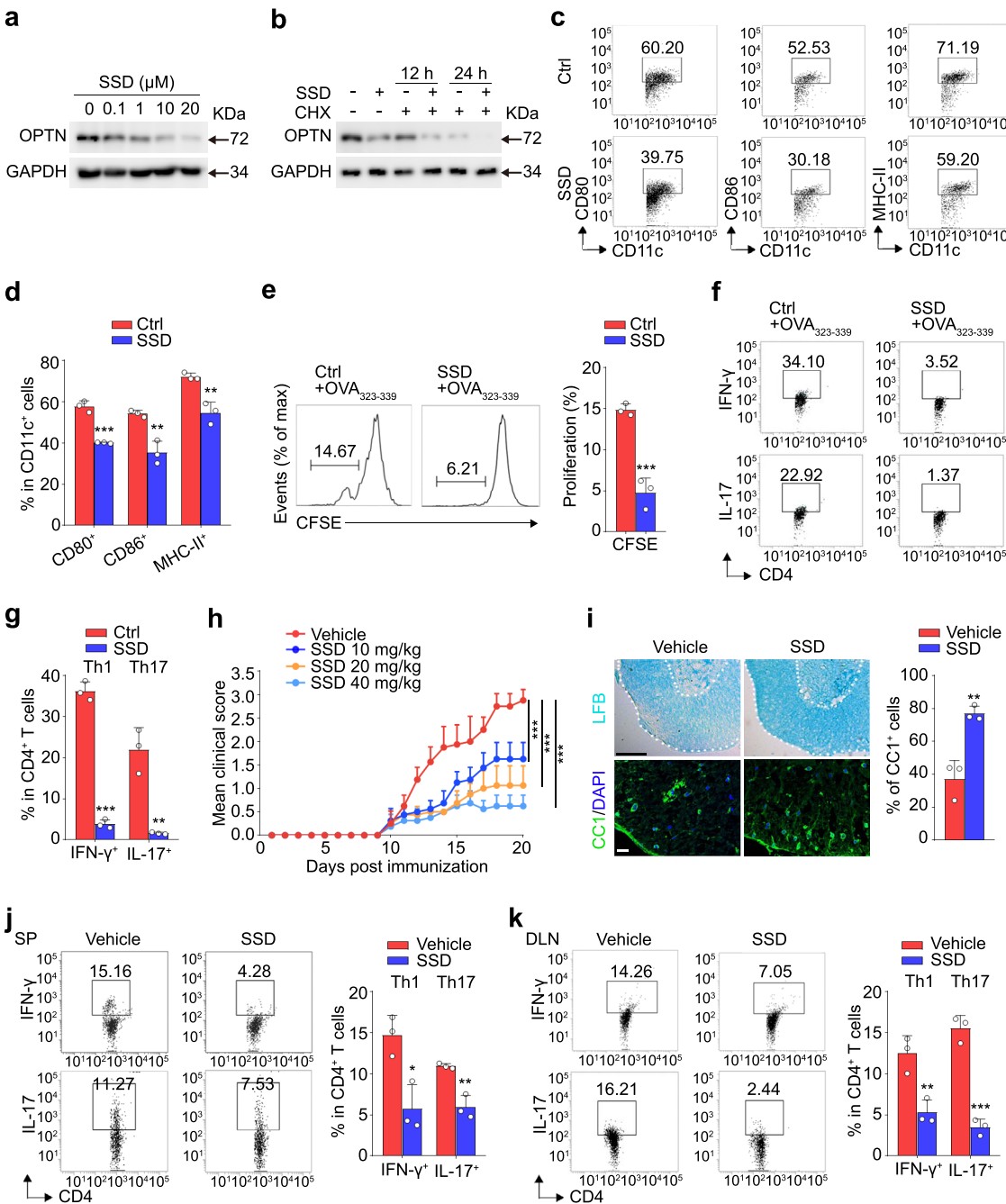

**Fig. 8 OPTN inhibitor SSD impairs DC function and attenuates EAE. a** Western blotting analysis of BMDCs that treated with SSD of different concentrations for 24 h before LPS stimulation. $n = 3$ independent experiments. **b** Western blotting analysis of 293T cells that treated with cycloheximide (CHX) for different time after SSD (1 μM) treatment. $n = 3$ independent experiments. **c, d** BMDCs were cultured with SSD (1 μM) for 24 h before LPS stimulation. Flow cytometry analysis for the expression of CD80, CD86, and MHC-II. $n = 3$ independent experiments, $P_{(CD80)} = 0.0004$, $P_{(CD86)} = 0.0049$, $P_{(MHC-II)} = 0.0050$. **e** Proliferation of CFSE-labeled OT-II CD4$^+$ T cells that incubated with BMDCs treated with or without SSD and then pulsed with OVA$_{323-339}$. $n = 3$ independent experiments, $P = 0.0009$. **f, g** Differentiation of OT-II CD4$^+$ T cells that incubated with BMDCs treated with or without SSD and then pulsed with OVA$_{323-339}$. $n = 3$ independent experiments, $P_{(IFN-\gamma)} = 2.6E-05$, $P_{(IL-17)} = 0.0028$. **h** C57BL/6 mice (8-week-old, female) were immunized with MOG$_{35-55}$ and administered daily with SSD or placebo solution intragastrically from the day of immunization. Mean clinical scores are shown. $n = 8$ independent animals, $P_{(SSD-10)} = 8.9E-08$, $P_{(SSD-20)} = 3.1E-11$, $P_{(SSD-40)} = 1.6E-13$. **i** (Upper) Luxolfast blue (LFB) staining of spinal cords from vehicle and SSD (40 mg/kg) treated EAE mice at day 20 after immunization. Scale bars, 200 μm. (Lower) Immunofluorescent labeling of CC1 (green) in the spinal cord of indicated mice at day 20 after immunization. Scale bars, 20 μm. $n = 3$ independent animals, $P = 0.0047$. **j, k** Flow cytometry analysis of Th1 and Th17 cells in spleen (SP) (**j**) and draining lymph nodes (DLN) (**k**) from vehicle and SSD treated EAE mice at day 20. $n = 3$ independent animals, $P_{(IFN-\gamma)} = 0.0148$, $P_{(IL-17)} = 0.0041$ in (**j**); $P_{(IFN-\gamma)} = 0.0095$, $P_{(IL-17)} = 0.0004$ in (**k**). Data are presented as means ± SD. *$P < 0.05$; **$P < 0.01$; ***$P < 0.001$. Unpaired two-tailed Student's $t$-test. Source data are provided in Source data file.

(Supplementary Fig. 3h–i). Thus, we can at least rule out the partial effects of *Optn* deletion in CD11c⁺ macrophages.

Then, our data showed that the expression of OPTN was augmented in both LPS stimulated MoDCs and BMDCs. The *Optn*-deficient DCs had lower levels of CD80, CD86, MHC-I, and MHC-II, along with impaired ability to promote CD4⁺ T-cell proliferation. In the light that the activated DCs, by TLR4 agonist LPS, could be used as a credible model system to dissect mechanisms in the TLR4 associated EAE[35,45,46], we employed EAE to further explore the biological role of DCs specific OPTN in regulating immune responses in vivo. As a result, we found that *CD11c*-Cre mediated deletion of *Optn* in CD11c⁺ cells protected mice from EAE by reducing the amount of pro-inflammatory Th1 and Th17 cells in the peripheral lymph system. Mechanistically, we found that OPTN did not interfere TLR4 or NF-κB pathway but negatively regulated the JAK2/STAT3 signaling pathway through direct interaction with the JH1 domain of JAK2 and subsequently *Il-10* transcription signals. Literature has reported that OPTN plays a critical role in selective autophagy and ubiquitination through its LC3 interaction region and Ub-binding domain[11]. However, our study showed that *Optn* ablation did not reverse the maturation of autophagy activated BMDCs, and OPTN without Ub-binding domain could promote DC maturation like OPTN but still retained the ability to bind to JAK2, indicating that the ubiquitination or autophagy function of OPTN does not participate in the regulation of JAK2/STAT3 signaling pathway in LPS stimulated BMDCs.

The process of DC maturation is facilitated by multiple signaling transductions, such as JAK-STAT3, NF-κB, MAPK, and Hippo-Mst pathways[47–49]. Although OPTN was reported to suppress TNF-mediated NF-κB activation[12], our data, however, demonstrated that *Optn* deficiency exerted no obvious effect on the NF-κB signaling in DCs, which was consistent with the previous report that *Optn*⁴⁷⁰ᵀ did not affect NF-κB activation in BMDCs[50]. Our results revealed that JAK2-STAT3 signaling pathways were strongly changed in LPS stimulated *Optn* deficient BMDCs. In parallel, it has been reported that cyclic helix B peptide inhibited TLR activation induced DC maturation by upregulating the suppressor of cytokine-signaling-1 (SOCS1) through JAK2-STAT3 signaling[51]. SOCS1 is the most potent JAK2 inhibitor through binding to the tyrosine residue (Y1007) in the activation loop of JH1 and recruiting the E3 ubiquitin ligase scaffold (Cullin5) to catalyze the ubiquitination/degradation of activated JAK2[52–54]. Here, we reported that OPTN also acted as a negative feedback inhibitor of JAK2 by directly binding to its JH1 domain. Moreover, such binding enabled OPTN to inhibit the interaction of STAT3 with JAK2. This finding accords with previous reports that TBK1, which could bind to and phosphorylated OPTN[55], directly phosphorylated STAT3 at serine 754 in the transactivation domain upon cytosolic DNA stimulation[56]. However, whether TBK1, OPTN, and JAK2 could form a complex is still unknown. Furthermore, consistent with the previous study showing that STAT3 deletion in DCs breaks immune tolerance and causes T-cell activation in mice, STAT3 inhibitor stattic rescued the expression of CD80 and CD86 in *Optn* deficient BMDCs[19,33,57].

Aberrant JAK2 activation induces abnormal haematopoiesis[58], which can nonetheless be utilized to ameliorate inflammatory diseases. For instance, JAK2-STAT3 activation is linked to improved cardiac remolding and reduced myocardial apoptosis in the metformin-treated type 2 diabetic model[59]. The phosphorylation of JAK2-STAT3 could also inhibit inflammation in non-alcoholic steatohepatitis[60]. Our study found that JAK2 activation, mediated by OPTN deprivation, effectively promoted the function of JAK2 in the immune system while suppressing DC activity.

It is worth noting that the relationship between TLRs and JAK2-STAT3 signaling pathway is complicated. On one hand, TLR regulates JAK/STAT pathway, as it has been reported that LPS treatment can activate JAK2 and STAT5[61]. On the other hand, STAT3 could downregulate TLR ligand-mediated gene expression, despite unclear mechanisms[62]. In the present study, we found that the levels of p-JAK2 and p-STAT3 were elevated 16 h after LPS stimulation (50 ng/mL), which may be related to the release of IL-6 and IL-10 by LPS-TLR4 pathway. In union, we conjectured that OPTN-regulated JAK2/STAT3 activation and DC maturation through the direct interaction with JAK2 instead of TLR4 signals.

Natural product SSD, with its anti-inflammatory and immune-modulatory properties, is one of the main bioactive components in traditional Chinese medicine, Bupleurum falcatum L[63]. Our data showed that SSD could effectively block OPTN expression through interfering with its stability, consistent with previous reports that certain compounds bound to a protein to induce its misfolding or degradation[64]. Furthermore, we found that SSD specifically depends on OPTN to modulate DC maturation and EAE progression, potentially representing an OPTN targeting drug for autoimmune disorders.

In conclusion, we demonstrated a critical role of OPTN in regulating DC maturation and hence autoimmune responses (Supplementary Fig. 8). Inhibition of OPTN via genetic down-regulation in CD11c⁺ cells can activate the JAK2-STAT3 signaling by extricating JAK2 and facilitating its dimerization, thus motivating the transcription function of STAT3, expediting the expression of *Il-10*, and finally impeding DC maturation and EAE injury. Together with the fact that pharmacological inhibition of OPTN by SSD can also prevent the progression of the experimental autoimmune disease, EAE, we highlight potentially promising DCs based immunotherapy for the treatment of autoimmune disease.

## Methods

**Mice.** C57BL/6 mice were obtained from Beijing Vital River Laboratory Animal Technology Co., Ltd (stock: 219). *CD11c*-Cre mice were acquired from the Jackson Laboratory (stock: 008068). OT-II mice were gifted by Prof. Lie Wang[65]. *Optn*ᶠˡ/ᶠˡ mice were gifted by Prof. Ronggui Hu[66]. The DCs specific *Optn*-null mice were generated by breeding *Optn*ᶠˡ/ᶠˡ mice with *CD11c*-Cre transgenic mice (*Optn*ᶠˡ/ᶠˡ; *CD11c*-Cre, *Optn*ΔCD11c). *Stat3*ᶠˡ/ᶠˡ mice were obtained from the Shanghai Model Organisms Center, Inc (stock: NM-CKO-200050), and intercrossed with *Optn*ΔCD11c mice to generate *Optn*ᶠˡ/ᶠˡ; *Stat3*ᶠˡ/ᶠˡ; *CD11c*-Cre (DKO) mice. All mice were housed in specific pathogen-free environment at 21 ± 1 °C and 60 ± 5% humidity, with a 12-h light/dark cycle. Experimental and control animals were bred separately. All mice were used at 6–8 weeks old, and euthanized with carbon dioxide. All animal use and studies were performed in compliance with all relevant ethical regulations, and were approved by the Institutional Animal Care and Use Committee (IACUC) at Zhejiang University.

**Human materials and MoDCs culture.** Blank human peripheral blood was collected from the Second Affiliated Hospital of School of Medicine, Zhejiang University. Samples from healthy people who showed no history of major diseases and had normal biochemical indicators and blood routine after physical examination were involved in this cohort. Volunteers provided informed consent for this study. All experiments were approved by the Human Subject Research Ethics Committee of the Second Affiliated Hospital of School of Medicine, Zhejiang University (No. Yan2016-003).

For human monocyte-derived-dendritic cells (MoDCs) culture, peripheral blood mononuclear cells (PBMCs) were isolated from blood through Ficoll-Hypaque (Mediatech Cellgro) density gradient centrifugation. Monocytes were purified from PBMCs using anti-CD14 microbeads (Miltenyi Biotech) and introduced into MoDCs in the presence of hGM-CSF (50 ng/mL; Peprotech) and hIL-4 (10 ng/mL; Peprotech). Half medium was replaced by fresh medium with GM-CSF and IL-4 at day 3 and day 5, and matured by LPS (300 ng/mL; Sigma-Aldrich) for 8 h at day 7.

**Cell isolation and culture.** For bone marrow-derived dendritic cells (BMDCs) preparation, bone marrow from 6 to 8-week-old C57BL/6 mice was cultured in RPMI-1640 medium (Gibco) supplemented with 10% fetal bovine serum (FBS)

(Gibco), recombinant mouse GM-CSF (50 ng/mL; Peprotech), and IL-4 (20 ng/mL; Peprotech). Half medium was replaced by fresh medium with GM-CSF and IL-4 at day 3, and used at day 6 as immature DCs. Immature DCs were stimulated with TLR agonists LPS (50 ng/mL; Sigma-Aldrich) or poly I:C (20 ng/mL; InvivoGen) for 8 h to generate mature DCs.

For bone marrow-derived macrophages (BMDMs) preparation, bone marrow from 6 to 8-week-old C57BL/6 mice was cultured in DMEM medium (Gibco) supplemented with 10% fetal bovine serum (FBS) (Gibco), recombinant mouse M-CSF (50 ng/mL; Peprotech). All supernate was replaced by fresh medium with M-CSF at day 3. BMDMs were then stimulated with LPS (50 ng/mL; Sigma-Aldrich) for M1 polarization, or IL-4 (20 ng/mL; Peprotech) for M2 polarization for 24 h at day 6.

CD11c$^+$ DCs were isolated from the spleen using CD11c positive selection kit (STEMCELL Technologies), and CD4$^+$ T cells were isolated from the lymph nodes using CD4 positive selection kit (STEMCELL Technologies).

**Cell transfection**. Transient transfection was performed using Lipofectamine 2000 (Invitrogen), according to the manufacturer's instructions, in 60-mm dishes with 2–3 µg of total DNA per transfection. HA-tagged-OPTN and OPTN plasmids were gifted from Pro. Ronggui Hu. FLAG-tagged-JAK1, FLAG-tagged-JAK2, and FLAG-tagged-JAK3 plasmids were obtained from OriGene. The truncation mutants of JAK2 were generated by PCR and subcloned into pCMV-Flag (Sigma-Aldrich). OPTN$^{\Delta UBD}$ plasmid was constructed by Genscript. The siRNAs of mouse OPTN (5′-CUGUUGUUUGAGAUGCAAATT-3′) were synthesized by Gene-Pharma. pLenti-GFP-mCherry-LC3B plasmid was obtained from Applied Biological Materials Inc.

**Drug screening**. HEK293 were cultured in DMEM medium (Gibco) supplemented with 10% FBS. For screening OPTN inhibitors, cells were treated with a series of natural products (10 µM; 24 h). The expression level of OPTN was analyzed by western blotting and quantified by Image J software (version 1.8.0).

**In vitro T-cell activation assay**. LPS activated BMDCs isolated from control and $Optn^{\Delta CD11c}$ mice were incubated with OVA$_{323–339}$ (10 µg/mL; Sangon) for 2 h and then incubated with $2 \times 10^5$ CFSE-labeled OT-II CD4$^+$ T cells (1:100) for 72 h. For MOG$_{35–55}$ stimulated T-cell activation assay, CD4$^+$ T cells isolated from control and $Optn^{\Delta CD11c}$ mice were co-incubated with DCs from control mice pulsed with MOG$_{35–55}$ peptide (20 µg/mL; Sangon) for 2 days. The proliferation and differentiation of CD4$^+$ T cells were measured by flow cytometry analysis.

**Induction and assessment of EAE**. To induce EAE, 8-week-old female C57BL/6 or $Optn^{\Delta CD11c}$ and DKO mice with their littermates were immunized subcutaneously with 400 µg MOG$_{35–55}$ peptide (Sangon) emulsified in CFA (Sigma) containing 4 mg/mL heat-killed Mycobacterium tuberculosis H37Ra (BD Biosciences) at day 0. In addition, 200 ng pertussis toxin (List Biological Laboratories) in 0.1 mL PBS was administered intravenously at day 0 and day 1 post immunization per mice. For adoptive transfer EAE, 8-week-old female C57BL/6 recipient mice were injected via the caudal vein with $5 \times 10^6$ MOG$_{35–55}$ specific control or $Optn^{\Delta CD11c}$ mice-derived CD4$^+$ enriched T cells in 0.1 mL PBS. For adoptive transfer of DCs to EAE mice, control or $Stat3^{fl/fl}$; $CD11c$-Cre BMDCs were stimulated with LPS (50 ng/mL) and MOG$_{35–55}$ (20 µg/mL) for 24 h, 8-week-old female C57BL/6 EAE mice were injected via the caudal vein with $2 \times 10^6$ BMDCs in 0.1 mL PBS for three times at day 1/3/7.

Animals were assessed daily for weight loss and clinical score system was defined as follows: 0, healthy; 0.5, partial paralysis of tail tip; 1, limp tail; 1.5, inhibition of hindlimbs; 2, weakness of hindlimbs; 2.5 hindlimb dragging; 3, one hindlimb paralysis; 3.5, complete hindlimb paralysis; 4, one forelimb paralysis; 4.5, complete forelimb paralysis; and 5, severe paralysis or death.

**Histological staining**. Spinal cords collected from PBS perfused EAE mice were fixed in 4% paraformaldehyde (PFA) overnight. For luxolfast blue (LFB) staining, tissues were embedded in OCT (SAKURA Finetek) and cut into 12-µm sections. Slides were stained with LFB solution (Sigma-Aldrich) overnight, rinsed with distilled water, and differentiated with 0.05% lithium carbonate solution followed by 70% ethanol. Differentiation was stopped by rinsing in distilled water.

Liver, lung, and spleen from control and $Optn^{\Delta CD11c}$ mice were fixed in 4% PFA overnight, then were embedded in paraffin and cut into 8-µm thick. Slides were deparaffinized and stained with hematoxylin and eosin (HE) for evaluation of inflammation.

**Immunofluorescence staining**. Spinal cord sections were permeabilized in PBS containing 0.5% Triton X-100 for 10 min at room temperature. Then primary antibodies of CC1 (Oncogene Research) were incubated at 4 °C overnight. Subsequently, slides were incubated with Alexa Fluor 488-coupled secondary antibodies (A-21202, Life Technologies, 1:1000). The nuclei were counterstained with DAPI (Dojindo). Images were taken with Leica SP8 microscope.

MoDCs were fixed with 4% PFA for 30 min at 4 °C and then permeabilized in PBS containing 0.5% Triton X-100 for 10 min at room temperature. Cells were then stained with primary antibodies to CD11c (Cell Signaling Technology) and OPTN (Santa Cruz) for 4 h. Subsequently, cells were incubated with Alexa Fluor 488-coupled secondary antibodies (A-11013, Life Technologies, 1:1000) or 568-coupled secondary antibodies (A-21090, Life Technologies, 1:1000). The nuclei were counterstained with DAPI. MoDCs and pLenti-GFP-mCherry-LC3B transfected BMDCs were imaged by Leica SP8 microscope.

**Flow cytometry**. The phenotypes of DCs and BMDMs were determined by flow cytometry. For cell surface staining, single-cell suspensions were incubated for 15 min at 4 °C with PE-anti-CD11c (117308, 0.2 µg/mL), FITC-anti-CD80 (104706, 0.2 µg/mL), PE/Cy7-anti-CD86 (105014, 0.2 µg/mL), FITC-anti-MHC-I (114606, 0.2 µg/mL), PerCP/Cy5.5-anti-MHC-II (107626, 0.2 µg/mL), FITC-anti-F4/80 (123108, 0.2 µg/mL), PerCP/Cy5.5-anti-CD11b (101228, 0.3 µg/mL), PE-anti-CD206 (141706, 0.2 µg/mL), APC-anti-CD115 (135509, 0.4 µg/mL), FITC-anti-PDCA-1 (127007, 0.3 µg/mL), PerCP/Cy5.5-anti-CD44 (103032, 0.2 µg/mL), PE-anti-CD62L (104407, 0.2 µg/mL) (all purchased from Biolegend). Samples were washed and then analyzed by FACS versus flow cytometry (BD Biosciences).

For intracellular staining of cytokines, single-cell suspensions from DLN and spleen of the indicated EAE mice were stained with FITC-anti-CD4 (553047, BD Biosciences, 0.2 µg/mL) first, followed by staining with PE-anti-IFN-γ (554412, BD Biosciences, 0.3 µg/mL), PE-anti-IL17A (506904, Biolegend, 0.3 µg/mL) and PE-anti-IL4 (12-7041-82, eBioScience, 0.4 µg/mL) using Cytofix/Cytoperm kit (BD Biosciences) according to the manufacturer's protocol. Intranuclear staining with PE-anti-Foxp3 (126404, Biolegend, 0.4 µg/mL) was performed using a Fixation/Permeabilization kit (eBioscience) according to the manufacturer's protocol.

**Western blotting**. Protein extracts were subjected to SDS-PAGE (8–12% gels) and blotted onto PVDF membranes. After blocking with 5% fat-free milk, the membranes were incubated with the following antibodies at 4 °C overnight: anti-OPTN (sc-166576, 1:1000), anti-TLR4 (sc-293072, 1:1000), and anti-Lamin B (sc-6216, 1:1000) were purchased from Santa Cruz; anti-JAK2 (#3230, 1:1000), anti-p-JAK2(Tyr1007/1008) (#3771, 1:1000), anti-STAT3 (#9139, 1:1000), anti-p-STAT3(Y705) (#9145, 1:1000), anti-MyD88 (#4283, 1:1000), anti-p-NF-κB (#3033, 1:1000), anti-HA (#3724, 1:1000), and anti-Flag (#86861, 1:1000) were purchased from Cell Signaling Technology; anti-NF-κB (ET1603-12, 1:1000), anti-p38 MAPK (ET1602-26, 1:1000), anti-p-p38 MAPK (ER1903-01, 1:1000), anti-IRF3 (ET1612-14, 1:1000), and anti-p-IRF3(S386) (ET1608-22, 1:1000) were purchased from Huabio; anti-GAPDH (db106, 1:5000) was purchased from DiagBio Technology. The bound antibodies were detected using horseradish peroxidase (HRP)-conjugated IgG (MULTI Sciences) and visualized with enhanced chemiluminescence (ECL, PerkinElmer) detection reagents (Thermo Scientific, USA). GAPDH or Lamin B was used as a loading control. Images were taken by GE AI600.

**Co-immunoprecipitation**. After transfection, HEK293 cells were lysed in lysis buffer (25 mM Tris (pH 8.0), 150 mM NaCl, 10% Glycerol, 1% NP40) supplemented with protease inhibitor cocktail. Whole-cell lysates were incubated with the indicated tag-beads for 2 h. The beads were washed using wash buffer (25 mM Tris (pH 8.0), 150 mM NaCl, 0.2% NP40) for at least five times, and then boiled in SDS loading buffer. Immunoprecipitated protein complexes were detected using western blotting. Anti-HA magnetic beads were bought from Bimake. Anti-HA and anti-Flag antibodies were bought from Santa Cruz. For co-immunoprecipitation of endogenous interaction between OPTN and JAK2, whole-cell lysates were incubated with OPTN (Santa Cruz) and control IgG (Santa Cruz), followed by immunoprecipitation using Protein A/G beads (Santa Cruz) and detection of JAK2 using JAK2-specific antibodies.

**RNA extraction and quantitative real-time PCR**. Total RNA was isolated from cells with Trizol reagent (Invitrogen), cDNA was transcribed using TransScript kit (TransGen Biotech). qRT-PCR analysis was performed using the SYBR Green (Bio-Rad) method on the ABI Fast 7500 real-time PCR instrument (Perkin-Elmer Applied Biosystems). The gene expression was normalized to the expression of the gene encoding $Gapdh$. Sequences of the primers for qRT-PCR are shown in Supplementary Table 1.

**ChIP-sequencing analysis**. We re-analyzed the published STAT3 ChIP-seq data in DCs from GEO dataset (GSE27161). The raw reads were processed through Trimmomatic v0.38[67] to cut the adapters with the default settings. Then we aligned the high-quality reads to the mm10 reference genome (from Ensembl, http://ftp.ensembl.org/pub/release-102/fasta/mus_musculus/dna/) using BWA v0.7.15 mem with the default settings[68]. Subsequently, peak calling was performed by MACS v1.4.1 with the following options '-g 1.87e+9 -s 38' and candidate peaks were further filtered with Enrichment ≥10-fold and $p \leq 10e{-}9$[69]. In addition, confident peaks were annotated by ChIP-seeker and Mus musculus gene files (from Ensembl, http://ftp.ensembl.org/pub/release-102/gff3/mus_musculus/)[70].

**Statistical analysis**. Flow cytometry analyses were performed by FlowJo (version 10.0.7 win64) and NovoExpress (version 1.5.0). ChIP-seq data were analyzed using Trimmomatic (v0.38), BWA (v0.7.15), MACS (v1.4.1), and IGV (version 2.8.13).

Statistical comparisons were performed using GraphPad Prism software (version 8.0.1) and Image J (version 1.8.0). Unpaired two-tailed Student's $t$-test or one-way ANOVA Tukey's post hoc analysis was performed to determine statistical significance between two samples or for multiple comparisons, respectively. All data are from at least three independent experiments and presented as means ± SD. The significance was considered to be when $p$ values were <0.05 (represented as *$p < 0.05$, **$p < 0.01$, ***$p < 0.001$, or not significant (NS)).

**Reporting summary**. Further information on research design is available in the Nature Research Reporting Summary linked to this article.

## Data availability

The LPS stimulated human MoDCs array analysis data downloaded and used in Fig. 1a are available in the NCBI Gene Expression Omnibus (GEO) database under accession code GSE2706. The STAT3 genome binding/occupancy profiling of splenic DCs data downloaded and used in Fig. 7j and Supplementary Fig. 6 are available in the NCBI GEO database accession code GSE27161. The reporting summary for this Article is available in the Supplementary Information file. All the other data supporting this study are available within this Article, Supplementary Information, Source data, or from the corresponding authors upon reasonable requests. Source data are provided with this paper.

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

## Acknowledgements

We acknowledge the financial support by the National Natural Science Foundation (Nos. 81872878, 82073857 to Q.J.W.), the Zhejiang Provincial Natural Science Foundation (No. LR21H310001 to Q.J.W., No. LGF19H310002 to Y.F.S.), the Medical and Health Science and Technology Program of Zhejiang Province (No. 2020KY054 to G.F.H.). We thank Prof. Lie Wang for gifted OT-II CD4+ T mice. We thank Prof. Ronggui Hu for gift of plasmids and *Optn*fl/fl mice. We thank Honghai Wu for breeding transgenic animals.

## Author contributions

Q.J.W. and J.J.W. conceived and designed the study. J.J.W., J.Y.W., W.X.H., L.L.Z., L.Q.S., C.Y.F., and Y.P.Q. performed the experiments. J.J.W., J.Y.W., W.X.H., Q.S., Y.F.S., L.J.Y., Z.X.Y., and J.C.W. performed the data analysis. J.J.W., J.Y.W., W.X.H., Q.J.W., B.Y., and Q.J.H. contributed to writing the manuscript. G.F.H. and J.C. gave some critical advices. Q.JW., B.Y., Q.J.H., Y.F.S., and G.F.H. contributed to the materials. All the authors read and approved the final version of the manuscript.

## Competing interests

The authors declare no competing interests.

## Additional information

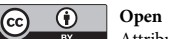

