## [Peer Review File · Nature Communications]

Reviewers' comments:

Reviewer #1 (JAK S/STAT signaling, MS-EAE)(Remarks to the Author):

In the current study, Weng et al found that DC-specific deficiency of OPTN impaired the abilities of DCs to prime CD4 T-cell and attenuated EAE through inhibiting the generation of IFN γ - and/or IL-17-producing CD4 T-cells. OPTN directly bound to JAK2, prevented JAK2-STAT3 interaction, and subsequent STAT3 phosphorylation. Deletion of STAT3 could increase the generation of inflammatory CD4 T cells and enhanced EAE in DC-specific Optn knockout mice. These findings are important and novel. However, the manuscript needs substantial improvement. My specific comments are:

1. Normally, LPS and polyI-C activate MyD88- and TRIF-dependent pathways. The authors found that OPTN directly bound to JAK2 and inhibited LPS- or polyI-C-induced JAK2 activation. How did TLRs activate the JAK/STAT pathway? How did the TLRs recruited JAK2 and STAT3?
2. In Fig 3G, GSEA analysis found that the genes of the STAT3 signaling pathway were enriched in Optn-deficient BMDCs. JAK2/STAT3 pathway activation was enhanced in the mutant DCs. Thus, the genes that were upregulated should be the targets of STAT3 but not the ones involved in STAT3 signaling.
3. What's the evidence showing that STAT3 could directly suppress the expression of CD80, CD86, MHC I and MHC II?
4. In Fig 4D, why were the molecular sizes of co-IPed flag-tagged JAK2 and JH1 proteins different from those of input proteins?
5. Could OPTN stimulation disrupt dimerization of endogenous JAK2 and the interaction between endogenous JAK2 and STAT3 in DCs?
6. The gating strategy for identifying IFN γ +, IL-17+, IL-4+, and FoxP3+ T cell were not clear. Naïve mice should be used as negative controls.
7. SSD bound to OPTN and inhibited its function. Why did SSD treatment result in the reduction of OPTN proteins (Fig 6A)?
8. If SSD could suppress EAE specifically through OPTN, what's the effect of SSD treatment on the functions of OPTN-deficient and OPTN/STAT3 double-deficient DCs and on EAE development in DC-specific OPTN knockout and OPTN/STAT3 DKO mice?
9. How many independent experiments were done in each figure?

Reviewer #2 (MS/EAE, neuro-immune crosstalk)(Remarks to the Author):

Unfortunately, I am unable to recommend this manuscript for publication. Without going into details, the paper is limited by general flaws which are listed in the following.

First of all, title, introduction and discussion are misleading. The thrust of the work is not on brain autoimmunity (EAE), but on optineurin's effect on DCs and their antigen presentation to CD4+ T cells. The authors describe optineurin as a multifunctional protein "involved in several immune diseases including CD and Paget's". The motivation, why optineurin was explored in MS/EAE remains obscure.

Is there any known connection between the molecule and MS, for example by GWAS? EAE seems to be used as a model to verify in vitro results, which as a routine is done by many immunologists. The overall message is meager. Like many gene depletion studies, optineurin KO mice show a partially decreased response to active EAE induction, which may be due to an impact on peripheral activation. This is not really borne out, and effector mechanisms also remain largely unexplored. Thus, it appears bold to predict optineurin as a possible target in clinical MS therapy.

Reviewer #3 (DC signaling/biology)(Remarks to the Author):

Weng et al.

The study by Weng et al. describes a role for Optn in LPS-induced DC maturation and priming of CD4 T cells. The deletion of Optn in DC in vivo inhibits disease progression in the mouse EAE model. The authors also identify an inhibitor of LPS-induced Optn expression that also inhibits disease progression in the EAE model and CD4 T cell activation. Mechanistically, the authors show increased JAK-STAT activation in Optn deficient DC and suggest Optn-mediated negative regulation of STAT3 leads to increased DC maturation and CD4 T cell activation.

There are several interesting observations in the study, but also a number of issues with the interpretation of the results in light of the published literature that should be addressed. Firstly, there is no mention of the known functions of Optn and how they may relate to the findings. In particular, the role of ubiquitin-binding, Optn is one of 5 unique proteins in the mammalian genome that possess a UBAN domain capable of binding both K63-linked and linear ubiquitin chains that are critical to TLR signaling and a number of other innate immune signaling pathways. How ub-binding relates to the functions of Optn in JAK-STAT signaling is a critical issue. Another well-characterised function of Optn is in selective autophagy, acting as a cargo receptor by virtue of both its UBAN domain and an LC3 interacting motif. Given the important role of autophagy is the negative regulation of innate immune signaling pathways through the degradation of cytoplasmic signaling complexes, the implications of these functions in the role of Optn in DC maturation is also a critical issue.

Specific points to be addressed:

- 1) In reference to fig.1b the authors state p62 and Optn were the "most upregulated genes" in LPS stimulated mouse BMDC, what is the relative level of increased p62 and Optn expression compared with other genes associated with DC maturation such as CD40, cytokines (TNF α , IL-1 β) and chemokines (Cxcl10, Cxcl9). These do not appear in the heat map of differentially expressed genes or the qPCR analysis, despite normally being the most significantly induced genes in DC upon LPS stimulation.
- 2) Efficiency and specificity of Optn deletion in Optn^{f/f}.Cd11c-cre mice needs to be confirmed both in vivo and with BMDC in vitro. This is critical since the Cd11c-cre line has been shown to have problems with germline recombination, and also off target expression in T cells. Previous reports have also shown poor deletion efficiency in BMDC in vitro with this line.
- 3) The full gating strategy for spleen and LN DC is required.
- 4) Analysis of resident vs migratory DC suggests a proportionally equal reduction, the reduction of total DC being most significant. Total numbers should be shown as well as % (this goes for all facts

data).

5) In Fig.1I, FMO should be used to gate positive staining for IFN γ (same goes for cytokine analysis in fig.2).

6) Fig.3d shows GSEA analysis of up-regulated genes in Optn deficient DC, what are the down-regulated pathways and could they relate to the observed phenotype?

7) What is the relevance of LPS stimulated DC as a model system to dissect mechanisms in the EAE model? The adjuvant used doesn't even contain TLR4 ligands. This could be critical when considering cross-talk between pathways that may be activated in vivo that are not reflected in the in vitro experiments.

8) As mentioned above, there is no discussion of the Ub-binding properties of Optn and its role in autophagy. Optn specifically binds K63 and linear Ub chains generated during TLR signaling. Does the interaction between JAK and Optn require Ub binding? Is the interaction signal dependent?

9) Optn is a receptor for selective autophagy, which has been shown to down-regulate signaling pathways by degrading signaling complexes. Is autophagy involved in the role of Optn in regulating JAK signaling?

10) Fig.3L rescue experiment is not convincing. The authors should measure costimulatory molecule expression by qPCR to validate these experiments.

11) Fig.5A; Stat3^{f/f}.Cd11c-cre mice alone should be shown as a control. Deletion efficiency and specificity for both Optn and Stat3 in DKO mice should be shown.

12) Fig.6; SSD inhibits Optn expression (not activity), does it act as an antagonist of TLR signaling?

13) Fig.S1; full gating strategy needed for both LN and spleen.

14) Fig.S4 needs total kinase blots as loading control for phospho-kinases. What were the pPI3K and pNF- κ B antibodies used?

Comments and Responses (NCOMMS-20-02776A)

Responses to the comments of Reviewer 1:

Reviewer #1 (JAK S/STAT signaling, MS-EAE)(Remarks to the Author):

In the current study, Weng et al found that DC-specific deficiency of OPTN impaired the abilities of DCs to prime CD4 T-cell and attenuated EAE through inhibiting the generation of IFN γ - and/or IL-17-producing CD4 T-cells. OPTN directly bound to JAK2, prevented JAK2-STAT3 interaction, and subsequent STAT3 phosphorylation. Deletion of STAT3 could increase the generation of inflammatory CD4 T cells and enhanced EAE in DC-specific Optn knockout mice. These findings are important and novel. However, the manuscript needs substantial improvement. My specific comments are:

Response: We appreciate the reviewer's comments on the significance of our work. We have performed additional experiments and revised the manuscript according to the comments.

1. Normally, LPS and poly I-C activate MyD88- and TRIF -dependent pathways. The authors found that OPTN directly bound to JAK2 and inhibited LPS- or polyI-C-induced JAK2 activation. How did TLRs activate the JAK/STAT pathway? How did the TLRs recruited JAK2 and STAT3?

Response: This is a good question. The relationship between TLRs and JAK/STAT signaling pathway is a complicated one. On one hand, TLR regulates JAK/STAT pathway, as it has been reported that LPS treatment can activate JAK2 and STAT5 while TLR-induced SOCS3 may, on the contrary, suppress STAT3 through IL-6 receptor^{1,2}. On the other hand, STAT3 could downregulate TLR-ligand-mediated gene expression, although the underlying mechanism remains unclear and it is believed that

certain unidentified STAT3 target gene(s) modulates(modulate) such downregulation³.

Our new western blotting data showed that in the control BMDCs, LPS stimulation increased the levels of TLR4 and MyD88, while p-JAK2 and p-STAT3 expressions remained unaffected (Fig. S9A, Columns 1 and 2). However, *Optn* deficiency attenuated the LPS-promoted increase of TLR4 and MyD88 levels while further augmented the p-JAK2 and p-STAT3 expressions (Fig. S9A, Columns 2 and 4).

Meanwhile, our study showed that OPTN interacts with the JH1 domain of JAK2. To find out whether TLRs are involved in this process, we performed an additional co-IP assay to detect the interaction between OPTN and TLR4 or MyD88 which is associate with the TIR domain of TLRs. The results showed that OPTN bound to MyD88, but not TLR4 (Fig. S9B). Thus, we speculate that TLR4, if involved in OPTN-mediated JAK/STAT pathway activation, may recruit JAK2/STAT3 through the MyD88-OPTN complex.

In conclusion, it is unclear at this point whether TLRs are involved in OPTN-mediated JAK/STAT pathway activation. It requires a comprehensive new study to explore this hypothesis and is beyond the scope of our present study. However, valuing this insight, we inserted our discussion in the revised manuscript, hoping to inspire further exploration (line 308-316).

2. In Fig 3G, GSEA analysis found that the genes of the STAT3 signaling pathway were enriched in Optn-deficient BMDCs. JAK2/STAT3 pathway activation was enhanced in the mutant DCs. Thus, the genes that were upregulated should be the targets of STAT3 but not the ones involved in STAT3 signaling.

Response: Thank you for pointing this out. We conducted a new GSEA analysis on the gene signatures of STAT3 targets in *Optn*^{ΔDC} BMDCs and replaced it with the old

figure. The new plot shows that the target genes of STAT3 were enriched in *Optn*-deficient BMDCs (Fig. 3G).

3. *What's the evidence showing that STAT3 could directly suppress the expression of CD80, CD86, MHC-I and MHC-II?*

Response: Thank you for this question. First of all, previous studies have revealed that *Stat3* ablation could elevate the expressions of CD80, CD86, MHC-II, and promote the maturation of CD11c⁺ DCs⁴. Also, STAT3 phosphorylation reportedly suppressed the gene transcription and protein production of CD80 and CD86⁵. In line with these prior findings, our results showed that the mRNA levels of *Cd80* and *Mhc-ii* (*Hl-dra*) in BMDCs were also increased by the treatment of stattic, a STAT3 inhibitor (Fig. 3M). In addition, qRT-PCR results showed that STAT3 overexpression significantly inhibited the mRNA level of *Cd80*, *Cd86*, *Mhc-i*, and *Mhc-ii* (Fig. 3N). Taken together, these results suggest that STAT3 could directly suppress the expression levels of CD80, CD86, MHC-I, and MHC-II.

4. *In Fig 4D, why were the molecular sizes of co-IPed flag-tagged JAK2 and JH1 proteins different from those of input proteins?*

Response: Sorry about this confusion. In the original experiment, the results from the input and co-IP samples were generated with gels of different concentrations (8% gel for input, 10% gel for co-IP samples), so the flag-tagged JAK2 and JH1 proteins appeared different. We have conducted a new experiment and replaced old data with new ones generated using the gels of the same concentration in figure 4D.

5. *Could OPTN stimulation disrupt dimerization of endogenous JAK2 and the interaction between endogenous JAK2 and STAT3 in DCs?*

Response: To answer this question, we conducted two additional experiments. First, new western blotting results showed that OPTN overexpression in BMDCs could disrupt the dimerization of JAK2 (Fig. 4J). We then performed co-IP and found that OPTN overexpression in BMDCs prevented the interaction between endogenous JAK2 and STAT3 (Fig. 4L).

6. *The gating strategy for identifying IFN γ ⁺, IL-17⁺, IL-4⁺, and FoxP3⁺ T cell were not clear. Naïve mice should be used as negative controls.*

Response: We appreciate this comment. To clarify, we provided the full gating strategy in the new Fig. S5. Data related to IFN γ ⁺, IL-17⁺, IL-4⁺, and FoxP3⁺ T cell identification are shown in Fig. S5E, F. Following the review's suggestion, we employed naïve mice as negative controls, as detailed in new Fig. S5I-L.

7. *SSD bound to OPTN and inhibited its function. Why did SSD treatment result in the reduction of OPTN proteins (Fig 6A)?*

Response: Thank you for this critical comment. Our original results showed that SSD could bind to OPTN protein and 10 μ M SSD could effectively reduce the *Optn* mRNA level and OPTN protein expression (old Fig. S5B, Fig. 6A-B). After careful contemplation, we believe these results might not be entirely valid.

(for reviewer only) Data 1. 10 μM SSD induced apoptosis in BMDCs. (A) BMDCs were treated with SSD (0.1, 1, 10 μM) for 24 h before LPS stimulation. Western blotting for expression of C-PARP and CC3. (B) BMDCs were cultured with SSD (0.1, 1, 10 μM) for 24 h before LPS stimulation. qRT-PCR analyses for mRNA level of *Optn*. Data are representative of three independent experiments, presented as means ± SEM. SSD-treated group compared to untreated group, ***, P < 0.001, Student's t-test.

To further clarify, we performed additional experiments and found that 10 μM SSD could cause apoptosis (Data 1A) and reduce the mRNA level of *Optn* (Data 1B) simultaneously in BMDCs. However, SSD of lower concentrations, such as 0.1 or 1 μM, exerted no effect on the *Optn* mRNA level (B), despite reducing the OPTN protein level. These results suggest that the decrease of *Optn* mRNA level caused by 10 μM SSD may be due to apoptosis, which accords with the previous finding that apoptotic cells exhibited lower mRNA levels⁶. To exclude the implication of apoptosis, we revised our experiment and employed SSD of 1 μM, instead of 10 μM, to explore its effect on OPTN expression and DC maturation. As a result, 1 μM SSD significantly inhibited OPTN expression in BMDCs (Fig.6A) and prevented BMDC maturation (Fig.6C-E).

We then proceeded to answer why SSD treatment resulted in the reduction of OPTN proteins. Prior studies showed that certain compounds could specifically bind to a protein and reduce its level by inducing protein misfolding⁷ or degradation⁸. To determine whether SSD affects the stability of OPTN protein, we treated BMDCs with 1 μM SSD and performed a cycloheximide (CHX) chase assay. As a result, the OPTN

protein level was decreased in the presence of 1 μ M SSD (Fig. 6B). These results suggest that the interaction of SSD with OPTN reduces the OPTN stability, which may result from protein misfolding after SSD treatment. Intrigued by this point, we included the related results and discussion in the updated manuscript (line317-323).

8. If SSD could suppress EAE specifically through OPTN, what's the effect of SSD treatment on the functions of OPTN-deficient and OPTN/STAT3 double-deficient DCs and on EAE development in DC-specific OPTN knockout and OPTN/STAT3 DKO mice?

Response: A series of new experiments were performed to address these questions. Firstly, we tested the effect of SSD treatment on the functions of *Optn*-deficient DCs. The qRT-PCR analyses of Ctrl and *Optn* ^{Δ DC} DCs were carried out with and without SSD administration. As a result, SSD treatment led to a similarly decreased mRNA levels of Cd80 and Cd86 in Ctrl and *Optn* ^{Δ DC} BMDC (Fig. S7F), indicating SSD exerts no significant effect on the functions of *Optn*-deficient DCs. Given that SSD suppresses OPTN expression, our result suggests that SSD mainly depends on OPTN to modulate DC functions.

Next, we explored whether SSD suppresses EAE through OPTN, by employing wildtype and OPTN-deficient EAE mice. The mice were administered daily with SSD (40 mg/kg) or placebo solution intragastrically from the day of immunization. The clinical score results showed that both SSD administration and OPTN deletion effectively inhibited the progression of EAE and SSD failed to further enhance the therapeutic effect in *Optn* ^{Δ DC} EAE mice (Fig. S7G). These results suggest that SSD specifically relies on OPTN to alleviate EAE.

To further confirm that SSD specifically targets OPTN, we employed an SPR study to analyze the interaction of SSD with OPTN or UBA3. As expected, the results showed that SSD bound OPTN (Fig. S7A, B), but not UBA3 (Fig. S7C, D).

Of note, we apologized that we did not conduct experiments on *Optn/Stat3* double-deficient DCs and mice as the number of *Optn/Stat3* double-deficient mice was insufficient due to the long breeding cycle.

9. *How many independent experiments were done in each figure?*

Response: We apologize for the omission of this information. At least three independent experiments were conducted for each experiment, and we have completed this information in each figure legend and Methods section.

Responses to the comments of Reviewer 2:

Reviewer #2 (MS/EAE, neuro-immune crosstalk)(Remarks to the Author):

1) Unfortunately, I am unable to recommend this manuscript for publication. Without going into details, the paper is limited by general flaws which are listed in the following. First of all, title, introduction and discussion are misleading. The thrust of the work is not on brain autoimmunity (EAE), but on optineurin's effect on DCs and their antigen presentation to CD4+ T cells.

Response: Thank you for making this valid point. We have revised the manuscript throughout, including the title, introduction, and discussion, to mainly focus on the function of OPTN on DC maturation and antigen presentation, rather than brain autoimmunity such as EAE. For instance, we changed the title to “Optineurin modulates the maturation of dendritic cells to regulate autoimmunity through JAK2-STAT3 signaling pathway”, modified the introduction to mainly describe the status and functions of DC maturation and the background of OPTN-related research, and re-organized our discussion.

2) The authors describe optineurin as a multifunctional protein “involved in several immune diseases including CD and Paget's”. The motivation, why optineurin was explored in MS/EAE remains obscure. Is there any known connection between the

molecule and MS, for example by GWAS? EAE seems to be used as a model to verify in vitro results, which as a routine is done by many immunologists.

Response: We apologize for not making clear our original research motive. The initial goal of our work is to study the cause of DC maturation and its functions. Coming across various literature associating OPTN with neuronal and immune diseases, we speculated that OPTN might play a role in regulating DC maturation and hence mediating the immune system. We employed EAE, the animal model of MS, simply to investigate the effect of OPTN on DCs and immune response *in vivo*. Similar approaches have been taken by other studies using EAE to determine the effect of certain gene deletion on the autoimmune modulation⁹. Interestingly, our results showed that OPTN-specific ablation in DCs can effectively ameliorate EAE development by inhibiting T-cell response. Inspired by this finding, we further analyzed the connection between OPTN and MS among published microarray database (MS brain lesions, GSE38010), and obtained affirmative results that the mRNA level of OPTN was increased in the sclerotic plaque of MS patients (Fig S4A), indicating that OPTN possibly participated in MS.

We hope this explains our approach of investigation from the beginning onward, which has also been clearly stated in the revised manuscript, to avoid further confusion.

3) The overall message is meager. Like many gene depletion studies, optineurin KO mice show a partially decreased response to active EAE induction, which may be due to an impact on peripheral activation. This is not really borne out, and effector mechanisms also remain largely unexplored.

Response: We understand that this comment is to question the effect of OPTN deletion on EAE and the underlying mechanism. First of all, we would like to restate that this study mainly investigates the effect of OPTN on DC functions, and EAE is used as an *in vivo* model for it.

Nonetheless, we provided in our manuscript various evidence that DC-specific OPTN deletion ameliorates EAE progression in mice through inhibiting autoimmune inflammation. For instance, OPTN knockout could lower the clinical score of EAE (Fig. 2A), alleviate demyelination (Fig. 2B), reduce MNC infiltration in the CNS (Fig. 2C), and inhibit T cell response in the peripheral tissues spleen and draining lymph nodes (Fig. 2D-G).

Mechanistically, we unveiled that DC-specific deficiency of OPTN impairs the abilities of DCs to prime CD4⁺ T-cell (Fig. 1) and attenuates EAE through inhibiting the generation of IFN γ - and/or IL-17-producing CD4⁺ T-cells (Fig. 2). OPTN directly binds to JAK2, prevents JAK2-STAT3 interaction and subsequent STAT3 phosphorylation (Fig. 3-4). The deletion of STAT3 could increase the generation of inflammatory CD4⁺ T-cells and enhance EAE in DC-specific *Optn* knockout mice (Fig. 5). The overall scheme is further illustrated in Fig. S10.

4) Thus, it appears bold to predict optineurin as a possible target in clinical MS therapy.

Response: Respecting the reviewer's comment, we avoided presenting OPTN as a possible target for MS treatment. Instead, we focused our manuscript more on the autoimmune modulation function of OPTN-mediated DC maturation.

Responses to the comments of Reviewer 3:

Reviewer #3 (DC signaling/biology)(Remarks to the Author):

The study by Weng et al. describes a role for Optn in LPS-induced DC maturation and priming of CD4 T cells. The deletion of Optn in DC in vivo inhibits disease progression in the mouse EAE model. The authors also identify and inhibitor of LPS-induced Optn expression that also inhibits disease progression in the EAE model and CD4 T cell activation. Mechanistically, the authors show increased JAK-STAT activation in Optn

deficient DC and suggest Optn-mediated negative regulation of STAT3 leads to increased DC maturation and CD4 T cell activation.

There are several interesting observations in the study, but also a number of issues with the interpretation of the results in light of the published literature that should be addressed.

Firstly, there is no mention of the known functions of Optn and how they may relate to the findings. In particular, the role of ubiquitin-binding, Optn is one of 5 unique proteins in the mammalian genome that possess a UBAN domain capable of binding both K63-linked and linear ubiquitin chains that are critical to TLR signaling and a number of other innate immune signaling pathways. How ub-binding relates to the functions of Optn in JAK-STAT signaling is a critical issue.

Another well-characterised function of Optn is in selective autophagy, acting as a cargo receptor by virtue of both its UBAN domain and an LC3 interacting motif. Given the important role of autophagy is the negative regulation of innate immune signaling pathways through the degradation of cytoplasmic signaling complexes, the implications of these functions in the role of Optn in DC maturation is also a critical issue.

Response: Thank you for your positive evaluation. In order to respond to the first part of the comment, we deleted the UBAN domain of OPTN and tested its interaction with JAK2. As a result, such deletion did not affect the ability of OPTN to bind to JAK2 (Fig. S8B), indicating ub-binding does not relate to the functions of OPTN in JAK-STAT signaling.

We then proceeded to address the second part of the comment. As we know, OPTN plays a critical role in selective autophagy by the virtue of both its LC3 interaction region and ubiquitin (Ub)-binding domain. We therefore tested the expressions of LC3 and ub-binding protein p62 to analyze the autophagy function of OPTN in DC maturation. The results showed comparable protein levels of LC3 and p62 in

LPS-treated Ctrl and *Optn*^{ΔDC} BMDCs (Fig. S8A), suggesting that OPTN-mediated autophagy did not participate in DC maturation.

Specific points to be addressed:

1) In reference to fig.1b the authors state p62 and Optn were the “most upregulated genes” in LPS stimulated mouse BMDC, what is the relative level of increased p62 and Optn expression compared with other genes associated with DC maturation such as CD40, cytokines (TNFa, IL-1b) and chemokines (Cxcl10, Cxcl9). These do not appear in the heat map of differentially expressed genes or the qPCR analysis, despite normally being the most significantly induced genes in DC upon LPS stimulation.

Response: Thank you for pointing this out. We newly added the genes associated with DC maturation into the heat map, including *CD40*, *TNFa*, *IL-1b*, *CXCL9*, and *CXCL10*. The data indicated they were differentially expressed (Fig. 1A). We then validated their expressions by qRT-PCR. The results showed that LPS stimulation significantly increased the levels of DC maturation markers (Fig. 1B). In particular, the increased levels of DC maturation-associated genes, such as *Cd40*, *Tnf*, *Il-1b*, *Cxcl10*, and *Cxcl9*, were higher than those of *p62* and *Optn*. Based on this finding, we revised the related statement in the manuscript to be more accurate (line76-79).

2) Efficiency and specificity of Optn deletion in Optn^Δ/Cd11c-cre mice need to be confirmed both in vivo and with BMDC in vitro. This is critical since the Cd11c-cre line has been shown to have problems with germline recombination, and also off target expression in T cells. Previous reports have also shown poor deletion efficiency in BMDC in vitro with this line.

Response: This is a valuable point. Following the reviewer's suggestion, we tested the efficiency and specificity of *Optn* deletion in the BMDCs, splenic CD11c⁺ DCs and CD4⁺ T-cells from the WT, *Optn*^{ΔDC}, and DKO mice. The western blotting data showed

that the protein levels of OPTN almost diminished in the BMDC and CD11c⁺ cells of *Optnf/f.Cd11c-cre* mice while only slightly decreased in CD4⁺ T cells (Fig.S1), suggesting that in our system, OPTN was effectively knocked out in DC lineage cells, while remained mostly stable in T-cells.

In addition, we carefully investigated the literature on DC-specific deletion of OPTN. DCs can usually express CD11c¹⁰ and CD11c-cre mice are hence often used to delete the floxed sequences in CD8⁻/CD8⁺ DCs, plasmacytoid DCs, or DCs derived from lymph nodes, lung, and epidermis (<https://www.jax.org/strain/008068>). Recently, it has been argued that CD11c does not provide sufficient DC specificity^{11,12}. For instance, the expression level of ITGA4 has also reportedly diminished in CD11c⁻ cells and lymphocyte subtypes. However, our test results showed a high specificity of *Optn* deletion in *Cd11c-cre* mice, which is necessary to validate our subsequent experimental results. Appreciating this point, we have included the relevant discussion in the updated manuscript (line 273-276).

3) The full gating strategy for spleen and LN DC is required.

Response: Thank you for reminding us. For the different DC types in spleen analyzed in Fig. 1J-K, the detailed gating strategies were added in Fig. S5A-B. Generally, we circled the primary cell population P1 first, and the CD11c (PE)⁺ MHC-II(PerCP-Cy5.5)⁺ cDCs and CD11c(PE)⁺ PDCA-1(FITC)⁺ pDCs were gated from P1.

For the different DC types in DLN analyzed in Fig. 1L, the detailed gating strategies were provided in Fig. S5C. We circled the primary cell population P2 first, the CD11c(PE)⁺ MHC-II(PerCP-Cy5.5)^{hi} M-DCs and CD11c(PE)⁺ MHC-II(PerCP-Cy5.5)^{low} R-DCs were gated from P2.

4) Analysis of resident vs migratory DC suggests a proportionally equal reduction, the reduction of total DC being most significant. Total numbers should be shown as well as % (this goes for all facs data).

Response: Thank you for the comment. As advised, we added the total number and percentage of CD11c+ DCs in Fig. 1L. The results showed that the amount of total DCs also decreased in *Optn*^{ΔDC} mice, presumably due to the reduced M-DC level. The number of total DCs as % was also added in Fig. 1J-K.

5) *In Fig.1I, FMO should be used to gate positive staining for IFNγ (same goes for cytokine analysis in fig.2).*

Response: Thanks for the suggestion. We indeed used the FMO to gate positive staining for IFNγ and cytokine. The detailed FMO data are shown in Fig. S5D (for Fig.1I), Fig. S5E (for Fig.2D), and Fig. S5F (for Fig.2F).

6) *Fig.3d shows GSEA analysis of up-regulated genes in Optn deficient DC, what are the down-regulated pathways and could they relate to the observed phenotype?*

Response: This is an interesting question. We employed GSEA to reanalyze RNA-seq data, and the results showed that only the IL-12 pathway was down-regulated in *Optn*-deficient DCs (Fig. 3D). However, additional western blotting showed that *OPTN* deletion did not affect the phosphorylation of JUN, a downstream effector of the IL-12 pathway, in *Optn*^{ΔDC} BMDCs (Fig. S6), suggesting that the downregulation of IL-12 pathway is not involved in *OPTN*-mediated DC maturation.

(for reviewer only) **Data2. DC-related signaling pathway in OPTN deficient BMDCs.** mRNA was isolated from LPS-stimulated Ctrl or *Optn*^{ΔDC} BMDCs, and subjected to RNA-seq analysis. (A-F) Indicated signaling pathway were analyzed by GSEA.

Moreover, we analyzed other DC-related signaling pathways, including CREB, ERK, HIPPO, PTEN, JNK, and GSK3. The results show that none of them significantly changed upon OPTN deletion (Data 2).

7) *What is the relevance of LPS stimulated DC as a model system to dissect mechanisms in the EAE model? The adjuvant used doesn't even contain TLR4 ligands. This could be critical when considering cross-talk between pathways that may be activated in vivo that are not reflected in the in vitro experiments.*

Response: Thank you for the question. In our study, we used MOG₃₅₋₅₅, complete Freund's adjuvant, PTX, and Mycobacterium tuberculosis H37Ra to induce EAE. PTX could facilitate immune cell entering the CNS by increasing permeability across the blood-brain barrier, and studies have indicated that PTX-induced leukocyte recruitment in EAE mice depends on TLR4¹³. PTX also has the ability to promote DC maturation. PTX-treated DC transfection induced EAE in mice¹⁴. Moreover, Mycobacterium

tuberculosis could increase the virulence and immunogenicity of H37Ra by the early secretory antigenic target protein 6 (ESAT6)¹⁵, a virulent factor that could drive the activation and maturation of BMDCs via TLR4-mediated signaling¹⁶. Taken together, these findings indicate that DCs in EAE mice can be activated by either PTX or *Mycobacterium tuberculosis* through TLR4. Thus, DCs stimulated by LPS, a TLR4 agonist, can be used as a credible model system to dissect mechanisms in the TLR4-associated EAE model. We explained this point in the Discussion of the revised manuscript as well (line 268-271).

8) As mentioned above, there is no discussion of the ub-binding properties of Optn and its role in autophagy. Optn specifically binds K63 and linear Ub chains generated during TLR signaling. Does the interaction between JAK and Optn require Ub binding? Is the interaction signal dependent?

Response: Thank you for emphasizing this crucial point. We have addressed this issue in the very first portion of our response to the reviewer. Again, the answer is the JAK-OPTN interaction does not involve ub-binding.

9) Optn is a receptor for selective autophagy, which has been shown to down-regulate signaling pathways by degrading signaling complexes. Is autophagy involved in the role of Optn in regulating JAK signaling?

Response: To further confirm whether autophagy is involved in the OPTN-regulated JAK signaling pathway, we tested the expression levels of p-JAK2, p-STAT3, p62, and LC3 in Ctrl and Optn^{ΔDC} BMDCs after LPS stimulation. In line with the original data (Fig. 3E, F), the p-JAK2 and p-STAT3 levels were increased in Optn^{ΔDC} BMDCs while the p62 and LC3 expressions remained stable with or without LPS stimulation (Fig. S8A), indicating autophagy is not involved in *Optn*-regulating JAK signaling. Moreover, Fig. S8A also shows that suppressing autophagy by Chloroquine (CQ) could significantly inhibit the expressions of p-STAT3 and p-JAK2 in both control and

OPTN^{ΔDC} BMDCs, and OPTN^{ΔDC} BMDCs showing higher p-STAT3 and p-JAK2 levels than Ctrl BMDCs. These results reinforced that the autophagy does not participate in the OPTN-mediated JAK signaling pathway under LPS treatment. The topic is discussed in the updated manuscript as well (line 277-282).

10) Fig.3L rescue experiment is not convincing. The authors should measure costimulatory molecule expression by qPCR to validate these experiments.

Response: Thank you for pointing this out. Following the advice, we treated Ctrl and Optn^{ΔDC} BMDCs with static before LPS stimulation, and analyzed the mRNA levels of *Cd86* and *Hl-dra* (MHC-II) molecules by qRT-PCR. The results showed that static treatment restored the mRNA level of *Cd86* and *Hl-dra* in LPS-stimulated Optn^{ΔDC} BMDCs (Fig 3M), which is consistent with the FACS results in Fig. 3L.

11) Fig.5A; Stat3^{f/f}.Cd11c-cre mice alone should be shown as a control. Deletion efficiency and specificity for both Optn and Stat3 in DKO mice should be shown.

Response: Thank you for this suggestion. However, the number of Stat3^{f/f}.Cd11c-cre mice was insufficient to be included regretfully, due to the long breeding cycle.

To validate the efficiency and specificity of OPTN and STAT3, we performed western blotting on the BMDCs, splenic CD11c⁺ DCs and lymph node CD4⁺ T cells from the WT or DKO mice. The results showed that protein levels of both OPTN and STAT3 almost diminished in the BMDCs and CD11c⁺ cells of DKO mice, while only slightly decreased in CD4⁺ T cells (Fig.S1), suggesting that in our system, OPTN and STAT3 could be efficiently and specifically knocked out in DC lineage cells.

12) Fig.6; SSD inhibits Optn expression (not activity), does it act as an antagonist of TLR signaling?

Response: Thanks for the question. To find the answer, we performed western blotting and the results showed that SSD treatment exerted no effect on the expression levels of TLR4 and Myd88 in LPS-treated BMDCs (Fig. S7E), suggesting that SSD is not the antagonist of TLR signaling. We also discussed this point in the updated manuscript. (line 247-249).

13) Fig.S1; full gating strategy needed for both LN and spleen.

Response: Thanks for the emphasis. In figure S1A (revised figure S2A), we analyzed the percentage of CD11c (PE) positive DCs in bone marrow of Ctrl and Optn^{ΔDC} mice, P1 is main cell population circled in FSC-SSC diagram, CD11c+ DCs (P4) were gated from P1. The data for full gating strategy were shown in Fig. S5G (for original Fig. 1A) and Fig. S5H (for original Fig. 1B).

14) Fig.S4 needs total kinase blots as loading control for phospho-kinases. What were the pPI3K and pNF-κB antibodies used?

Response: Thank you for pointing this out. We re-collected the samples and analyzed the expression levels of the proteins and their phosphorylated forms, as presented in the revised Fig.S6. The antibodies for pPI3K and pNF-κB were p-PI3K (CST, #4228) and Phospho-NF-κB (CST, #3033), respectively, which are also listed in the Methods.

References

- 1 Kimura, A. *et al.* Suppressor of cytokine signaling-1 selectively inhibits LPS-induced IL-6 production by regulating JAK-STAT. *Proc Natl Acad Sci U S A* **102**, 17089-17094 (2005).
- 2 Yasukawa, H. *et al.* IL-6 induces an anti-inflammatory response in the absence of SOCS3 in macrophages. *Nat. Immunol.* **4**, 551-556 (2003).
- 3 Murray, P. J. The JAK-STAT signaling pathway: input and output integration. *J Immunol* **178**, 2623-2629 (2007).
- 4 Kortylewski, M. *et al.* Toll-like receptor 9 activation of signal transducer and activator of transcription 3 constrains its agonist-based immunotherapy. *Cancer Res.* **69**, 2497-2505 (2009).
- 5 Kowalczyk, A., D'Souza, C. A. & Zhang, L. Cell-extrinsic CTLA4-mediated regulation of dendritic cell maturation depends on STAT3. *Eur J Immunol* **44**, 1143-1155 (2014).
- 6 Del Prete, M. J. *et al.* Degradation of cellular mRNA is a general early apoptosis-induced event. *FASEB J.* **16**, 2003-2005 (2002).
- 7 Lebraud, H. & Heightman, T. D. Protein degradation: a validated therapeutic strategy with exciting prospects. *Essays Biochem.* **61**, 517-527 (2017).
- 8 Winter, G. E. *et al.* DRUG DEVELOPMENT. Phthalimide conjugation as a strategy for in vivo target protein degradation. *Science* **348**, 1376-1381 (2015).
- 9 Quan, M. Y. *et al.* Amlexanox attenuates experimental autoimmune encephalomyelitis by inhibiting dendritic cell maturation and reprogramming effector and regulatory T cell responses. *J. Neuroinflammation* **16**, 52 (2019).
- 10 Liu, J. & Cao, X. Regulatory dendritic cells in autoimmunity: A comprehensive review. *J. Autoimmun.* **63**, 1-12 (2015).
- 11 Brown, C. C. *et al.* Transcriptional Basis of Mouse and Human Dendritic Cell Heterogeneity. *Cell* **179**, 846-863 e824 (2019).
- 12 Murphy, K. M. Comment on "Activation of β -catenin in dendritic cells regulates immunity versus tolerance in the intestine". *Science* **333**, 405; author reply 405 (2011).
- 13 Kerfoot, S. M. *et al.* TLR4 contributes to disease-inducing mechanisms resulting in central nervous system autoimmune disease. *J Immunol* **173**, 7070-7077 (2004).
- 14 Zhou, H. *et al.* Differential IL-10 production by DCs determines the distinct adjuvant effects of LPS and PTX in EAE induction. *Eur J Immunol* **44**, 1352-1362 (2014).
- 15 Guo, S. *et al.* The CFP10/ESAT6 complex of Mycobacterium tuberculosis may function as a regulator of macrophage cell death at different stages of tuberculosis infection. *Med. Hypotheses* **78**, 389-392 (2012).
- 16 Jang, A. R. *et al.* Mycobacterium tuberculosis ESAT6 Drives the Activation and Maturation of Bone Marrow-Derived Dendritic Cells via TLR4-Mediated Signaling. *Immune Netw.* **19**, e13 (2019).

REVIEWER COMMENTS

Reviewer #1 (Remarks to the Author):

The revised addressed the majority of my questions but failed to address some questions.

1. Normally, LPS and polyI-C activate MyD88- and TRIF-dependent pathways. The authors found that OPTN directly bound to JAK2 and inhibited LPS- or polyI-C-induced JAK2 activation. How did TLRs activate the JAK/STAT pathway? How did the TLRs recruited JAK2 and STAT3?

Knowing how the TLRs recruited JAK2 and STAT3 to activate the JAK/STAT pathway is important for understanding the molecular mechanism underlying the findings of this manuscript. The revised manuscript did not fully address this question. In contrast, the data (Fig S9A) from the revised manuscript clearly demonstrated that LPS did not activate JAK2 or Stat3.

2. In Fig 3G, GSEA analysis found that the genes of the STAT3 signaling pathway were enriched in Optn-deficient BMDCs. JAK2/STAT3 pathway activation was enhanced in the mutant DCs. Thus, the genes that were upregulated should be the targets of STAT3 but not the ones involved in STAT3 signaling.

The authors provided the new data showing that the target genes of STAT3 were enriched in Optn-deficient BMDCs (Fig. 3G). However, the detailed information/citation about these STAT3 target genes was missing.

3. What's the evidence showing that STAT3 could directly suppress the expression of CD80, CD86, MHC I and MHC II?

The authors showed that STAT3 deficiency/reduction increased and STAT3 overexpression decreased the mRNA level of Cd80, Cd86, Mhc-i, and Mhc-ii, respectively. However, they failed to provide evidences showing STAT3 directly suppressed the expression of these target genes through binding to the gene regulatory sites.

Reviewer #3 (Remarks to the Author):

The authors present new experiments to address the role of the Ub-binding domain of Optn in the interaction with Stat3 and correlation of autophagy with the observed phenotype. However, I think these experiments are too preliminary to draw the conclusion that neither Ub-binding or autophagy play a role. For example, what is the impact of the UBAN mutation in Optn on DC activation? If similar effects to Optn deletion are observed, the conclusion should be that the Stat3 binding is redundant in this phenotype. Similarly, what is the effect of autophagy deficiency on Stat3 and DC activation? Selective autophagy or slight changes in autophagic flux may not be detected by the crude LC3 immunoblot assays shown, but could have an important impact on signaling.

Regarding point 2) Efficiency and specificity of Optn deletion.

The new data There shows a significant deletion of Optn in CD4+ T cells, this could impact the interpretation of the in vivo results. To rule out a role for Optn in CD4+ T cells in the in vivo phenotype, the EAE model is driven by CD4+ T cell activation so it's critical to rule out an intrinsic impact of Optn deletion in these cells on disease progression. This could be done with mixed chimera

experiments, for example.

Regarding point 11) Stat3^{f/f}.Cd11c-cre mice.

The phenotype in Stat3^{f/f}.Cd11c-cre mice is critical to the interpretation of the data. Does Stat3 in DC have a role in limiting EAE, or is this only seen upon Optn deletion? The characterization of these mice is important, several studies have shown the important role of Stat3 in macrophages for limiting inflammation (due to IL-10 signaling), since Cd11c-cre also targets macrophages, can the authors rule out a role for Stat3 in these cells mitigating the effects of Optn deletion in DC?

Regarding point 12) Does it act as an antagonist of TLR signaling?

Why question was; does SSD affect TLR signaling? i.e., activation of MAPK, NF- κ B and IRF3 pathways. Not expression of TLR4 or MyD88.

Point-to-point responses to Reviewers' concerns:

Reviewer #1(Remarks to the Author):

The revised addressed the majority of my questions but failed to address some questions.

1. Normally, LPS and polyI-C activate MyD88- and TRIF-dependent pathways. The authors found that OPTN directly bound to JAK2 and inhibited LPS- or polyI-C-induced JAK2 activation. How did TLRs activate the JAK/STAT pathway? How did the TLRs recruited JAK2 and STAT3?

Knowing how the TLRs recruited JAK2 and STAT3 to activate the JAK/STAT pathway is important for understanding the molecular mechanism underlying the findings of this manuscript. The revised manuscript did not fully address this question. In contrast, the data (Fig S9A) from the revised manuscript clearly demonstrated that LPS did not activate JAK2 or Stat3.

Response: Thank you for your critical comments. LPS has been shown to activate JAK2 and STAT3 signaling in dendritic cells in previous studies^{1,2}. Consistently, our study found JAK2/STAT3 pathway was activated in WT BMDCs after LPS treatment for 16 h rather than 8 h (revised Fig. 3a, previous Fig S9A also showed no activation for 8 h of LPS treatment), however, it was enough for JAK2/STAT3 activation in *Optn* knockout BMDCs after 8 h of LPS treatment (revised Fig. 3b).

Given that IL-6 and IL-10, classical stimulators of JAK2, are rapid released and then go down in response to LPS stimulation³⁻⁵, JAK2 can bind to these cytokine receptors and be activated by increased cytokines like IL-6 and IL-10⁶. Our results revealed that both IL-6 and IL-10 antibodies blocked the phosphorylated activation of JAK2 and STAT3 in BMDCs (revised Fig. 4d), while co-IP results showed that TLR4 (the receptor of LPS) cannot directly bind with JAK2 or STAT3 in BMDCs (revised Fig. 4e), indicating that it is IL-6/IL-10 receptors rather than TLRs to directly recruit JAK2 to activate JAK2/STAT3 pathway in BMDCs. Our study further found that IL-10 was the direct target of STAT3 in BMDCs, and *Optn* knockout activated STAT3 and then increased and maintained the expression of IL-10 but not IL-6 (revised Fig. 5j, k), exaggerating the function of IL-10 and resulting in the evident activation of JAK2/STAT3 pathway.

Taken together, we concluded that TLR4 may control JAK2/STAT3 activation through IL-6 and IL-10 signal in BMDCs. *Optn* deletion upregulated JAK2/STAT3 phosphorylation and accumulated IL-10 cytokine expression, thus promoting IL-10/JAK2/STAT3/IL-10 loop and forming a positive feedback, finally blocking DC maturation. We have included these new data in revised Fig. 3a, b; Fig. 4d, e and Fig. 5j,

k, and discussed this issue in the revised text (page 7, line 15 to 21; page 9, line 22 to page 10, line 2 and page 11, line 7 to line 21).

2. In Fig 3G, GSEA analysis found that the genes of the STAT3 signaling pathway were enriched in *Optn*-deficient BMDCs. JAK2/STAT3 pathway activation was enhanced in the mutant DCs. Thus, the genes that were upregulated should be the targets of STAT3 but not the ones involved in STAT3 signaling.

The authors provided the new data showing that the target genes of STAT3 were enriched in *Optn*-deficient BMDCs (Fig. 3G). However, the detailed information /citation about these STAT3 target genes was missing.

Response: Thank you for your critical suggestions. We agree with the reviewer that the genes involved in STAT3 signaling pathway should not be changed and enriched in *Optn* knockout DCs. Since we only did once for RNA-seq analysis with no biological replication, it is possible that we got the false-positive data. To make the data in this paper more accurate and rigorous, we just deleted all analysed RNA-seq data. Instead, we added more experimental data to show the activated JAK2/STAT3 pathway and increased expression of STAT3 target genes in *Optn* deficient BMDCs (revised Fig. 3d, g, 4d, 5j, k). Especially, the target genes of STAT3, including like *Vegfa*, *Hif1a*, *Hgf*, *Il-10*, *Il-13* and so on has been verified in previous studies⁷⁻¹⁰, which were also discussed and cited in the revised text (page 7, line 21 to 28; page 8, line 2 to 4; page 9, line 22 to 26 and page 11, line 7 to line 21).

3. What's the evidence showing that STAT3 could directly suppress the expression of CD80, CD86, MHC I and MHC II?

The authors showed that STAT3 deficiency/reduction increased and STAT3 overexpression decreased the mRNA level of *Cd80*, *Cd86*, *Mhc-i*, and *Mhc-ii*, respectively. However, they failed to provide evidences showing STAT3 directly suppressed the expression of these target genes through binding to the gene regulatory sites.

Response: Thank you for your suggestion. To address the direct target genes of STAT3 in DCs, we re-analyzed the ChIP-seq data of STAT3 in silent and activated DCs from GSE27161¹¹. Results revealed that STAT3 had quite a high binding with *Il-10* gene in activated DCs (revised Fig. 5j), consistent with previous studies that STAT3 was a major transcription factor for *Il-10*^{10,12-14}. Compared with *Il-10* binding, STAT3 showed low or weak bindings with *Il-6*, *Cd80*, *Cd86*, *H2-K1* (*Mhc-i*) or *H2-Aa* (*Mhc-ii*) genes (revised Fig. 5j and revised Supplementary Fig. 5). Our study further found that *Optn* knockout in BMDCs increased and maintained the expression of IL-10 but not IL-6 (revised Fig. 5k), exaggerating the function of IL-10 and resulting in the evident activation of JAK2/STAT3.

These data indicated that STAT3 directly binds to *Il-10* gene and promote its transcription, then IL-10 may suppresses the expression of mature DC markers like *Cd80*, *Cd86*, *H2-K1* or *H2-Aa*¹⁵. Thus, our study showed that *Optn* deficiency facilitated the activation of JAK2/STAT3 pathway and subsequently induced *Il-10* transcription, which then restrained DC maturation via IL-10/JAK2/STAT3/IL-10 positive feedback loop. We have included the new data in revised Fig. 5j, k and Supplementary Fig. 5, and clarified this point in the revised text (page 11, line 7 to line 21).

Reviewer #3 (Remarks to the Author):

1. The authors present new experiments to address the role of the Ub-binding domain of Optn in the interaction with Stat3 and correlation of autophagy with the observed phenotype. However, I think these experiments are too preliminary to draw the conclusion that neither Ub-binding or autophagy play a role. For example, what is the impact of the UBAN mutation in Optn on DC activation? If similar effects to Optn deletion are observed, the conclusion should be that the Stat3 binding is redundant in this phenotype. Similarly, what is the effect of autophagy deficiency on Stat3 and DC activation? Selective autophagy or slight changes in autophagic flux may not be detected by the crude LC3 immunoblot assays shown, but could have an important impact on signaling.

Response: We thank the reviewer for the constructive suggestions. Results showed that OPTN^{ΔUBD} overexpression restored the expression of CD80 and CD86 in LPS-stimulated *Optn* deficient BMDCs, which had the similar effect to OPTN overexpression (revised Supplementary Fig. 2e, f). Besides, OPTN^{ΔUBD} is also able to interact with JAK2 (revised Fig. 5d), indicating that the function of OPTN on DC maturation and JAK2/STAT3 pathway activation is independent of its ubiquitin-binding domain.

Studies have found that the activation of STAT3 is closely related to autophagic block due to the transcriptional regulation, sequestration, and mitochondrial translocation of STAT3¹⁶⁻¹⁸. However, the specific effect of autophagy deficiency on STAT3 remains unclear in DCs. In our studies, treatment of RAPA didn't alter the level of p-JAK2 and p-STAT3 in *Optn*^{ΔDC} BMDCs (revised Fig. 4h). On the other hand, the role of autophagy on DC maturation and activation remains elusive. Although emerging literatures implicate autophagy is an essential mechanism in antigen loading to MHC II and antigen presentation by DCs^{19,20,21}, *Atg16L1*-deficient mice showed increased DC numbers and costimulatory molecule expression²². Especially, the induction of tolerogenic DC generation was associated with enhanced autophagy, and rapamycin (RAPA) activated autophagy can inhibit DC maturation and activation^{23,24}. Our results indicated that though *Optn* deletion inhibited the GFP⁺mCherry⁺ autophagic puncta, *Optn* deficiency cannot rescue the decrease of CD11c⁺CD80⁺ or CD11c⁺CD86⁺ cells by RAPA (revised Supplementary Fig.

2a-d). Together, these results suggested that OPTN-mediated autophagy did not contribute to the maturation defect in *Optn* knockout DCs.

We have included the new data in the revised Fig. 4h, 5d and Supplementary Fig. 2, and discussed this point in the revised text (page 8, line 8 to 18; page 10, line 7 to 9 and line 23 to 24).

2. Regarding point 2) Efficiency and specificity of Optn deletion.

The new data There shows a significant deletion of Optn in CD4+ T cells, this could impact the interpretation of the in vivo results. To rule out a role for Optn in CD4+ T cells in the in vivo phenotype, the EAE model is driven by CD4+ T cell activation so it's critical to rule out an intrinsic impact of Optn deletion in these cells on disease progression. This could be done with mixed chimera experiments, for example.

Response: We thank the reviewer for the suggestions. Exactly, *CD11c*-Cre line has been associated with germline recombination and off-target problem in T-cell²⁵. Our result also found a slight decrease of OPTN in CD4⁺ T cells from *Optn*^{ADC} mice (revised Supplementary Fig. 1a). Actually, results showed that the number of CD4⁺ and CD8⁺ T cells, the infiltration of lymphocytes in organs, and activation of CD4⁺ and CD8⁺ T cells in *Optn*^{ADC} mice were comparable with WT mice (revised Supplementary Fig. 1b-e), indicating that *CD11c*-conditional *Optn* knockout did not affect T-cell homeostasis. Besides, *in vitro* experiment also found the similar differentiation of CD4⁺ T-cells from WT and *Optn*^{ADC} mice after co-culture with WT BMDCs (revised Supplementary Fig. 1f).

To further rule out the role of *Optn*^{ADC} mice-derived CD4⁺ T cells in the *in vivo* phenotype of EAE mice, adoptive transfer of CD4⁺ T cells into EAE mice was performed²⁶⁻²⁸. Results indicated that *Optn*^{ADC} mice-derived CD4⁺ T cells had similar abilities to induce EAE when compared with WT mice-derived CD4⁺ T cells (revised Supplementary Fig. 1g-i). These data suggested that there is no impact of CD4⁺ T cells from *Optn*^{ADC} mice on the progression of EAE.

We have included the new data in the revised Supplementary Fig. 1, and discussed this point in the revised text (page 5, line 14 to 26 and page 7, line 1 to line 12).

3. Regarding point 11) Stat3ff.Cd11c-cre mice.

The phenotype in Stat3ff.Cd11c-cre mice is critical to the interpretation of the data. Does Stat3 in DC have a role in limiting EAE, or is this only seen upon Optn deletion? The characterization of these mice is important, several studies have shown the important role of Stat3 in macrophages for limiting inflammation (due to IL-10 signaling), since Cd11c-cre also targets macrophages, can the authors rule out a role for Stat3 in these cells mitigating the effects of Optn deletion in DC?

Response: We thank the reviewer for the suggestions. To identify whether STAT3 in DC has a role in limiting EAE, we re-generated *Stat3^{fl/fl}; CD11c-Cre* mice (*Stat3^{ADC}*, this mouse line died out during the Covid-19 expansion). Given that the requirement for EAE induction is quite restrict, like only 8-10 weeks old female C57 mice can be successfully induced, we were unable to generate many *Stat3^{ADC}* female mice at one time for EAE induction by now, so adoptive transfer experiment was applied to examine the role of *Stat3* cKO DCs during EAE development²⁹⁻³¹. STAT3 is a negative regulator for DC maturation³², our results further showed that adoptive transfer of BMDCs from *Stat3^{ADC}* mice aggravated EAE progression (revised Supplementary Fig. 3), indicating that STAT3 in DCs has the role of limiting EAE.

As the reviewer claimed, *CD11c-cre* also targets macrophages²⁵, and STAT3 in macrophages could limit the inflammation of EAE mice³³. To rule out the role of STAT3 in macrophages from *Optn^{ADC}* mice, our results showed that BMDMs from *Optn^{ADC}* and WT mice had similar levels of OPTN and (p-)STAT3 with or without IL-4 induction (revised Supplementary Fig. 4a). Moreover, BMDMs from WT and *Optn^{ADC}* mice had similar capacities on M1 polarization stimulated by LPS, M2 polarization induced by IL-4 and pro/anti-inflammatory cytokine expression (revised Supplementary Fig. 4b-e). In conclusion, *Optn* conditional knockout by *CD11c-Cre* has no effect on macrophages function.

We have included the new data in the revised Supplementary Fig. 3 and Supplementary Fig. 4, and discussed this point in the revised text (page 8, line 23 to 25 and page 9, line 5 to 14).

4. Regarding point 12) Does it act as an antagonist of TLR signaling?

Why question was; does SSD affect TLR signaling? i.e., activation of MAPK, NF-kB and IRF3 pathways. Not expression of TLR4 or MyD88.

Response: We thank the reviewer for pointing this out. We have provided new data showing that SSD did not affect TLR signaling, as indicated by no activation of MAPK, NF-kB and IRF3 pathways after SSD treatment in BMDCs (revised Supplementary Fig. 6e). We have included the new data in the revised Supplementary Fig. 6e, and discussed this point in the revised text (page 12, line 20 to 22).

Reference:

1. Okugawa, S., *et al.* Janus kinase 2 is involved in lipopolysaccharide-induced activation of macrophages. *Am J Physiol Cell Physiol* **285**, C399-408 (2003).
2. Lee, J.J., *et al.* Toll-like receptor 4-linked Janus kinase 2 signaling contributes to internalization of *Brucella abortus* by macrophages. *Infect Immun* **81**, 2448-2458 (2013).

3. Hobbs, S., Reynoso, M., Geddis, A.V., Mitrophanov, A.Y. & Matheny, R.W., Jr. LPS-stimulated NF-kappaB p65 dynamic response marks the initiation of TNF expression and transition to IL-10 expression in RAW 264.7 macrophages. *Physiol Rep* **6**, e13914 (2018).
4. Zhou, H., *et al.* Differential IL-10 production by DCs determines the distinct adjuvant effects of LPS and PTX in EAE induction. *Eur J Immunol* **44**, 1352-1362 (2014).
5. Perrin-Cocon, L., *et al.* TLR4 antagonist FP7 inhibits LPS-induced cytokine production and glycolytic reprogramming in dendritic cells, and protects mice from lethal influenza infection. *Sci Rep* **7**, 40791 (2017).
6. Leslie, L.A. & Younes, A. Targeting oncogenic and epigenetic survival pathways in lymphoma. *Leuk Lymphoma* **54**, 2365-2376 (2013).
7. Pham, T.H., *et al.* Inhibition of IL-13 and IL-13Ralpha2 Expression by IL-32theta in Human Monocytic Cells Requires PKCdelta and STAT3 Association. *Int J Mol Sci* **20**(2019).
8. Qin, J.J., Yan, L., Zhang, J. & Zhang, W.D. STAT3 as a potential therapeutic target in triple negative breast cancer: a systematic review. *J Exp Clin Cancer Res* **38**, 195 (2019).
9. Azare, J., *et al.* Constitutively activated Stat3 induces tumorigenesis and enhances cell motility of prostate epithelial cells through integrin beta 6. *Mol Cell Biol* **27**, 4444-4453 (2007).
10. Benkhart, E.M., Siedlar, M., Wedel, A., Werner, T. & Ziegler-Heitbrock, H.W. Role of Stat3 in lipopolysaccharide-induced IL-10 gene expression. *J Immunol* **165**, 1612-1617 (2000).
11. Wan, C.K., *et al.* The cytokines IL-21 and GM-CSF have opposing regulatory roles in the apoptosis of conventional dendritic cells. *Immunity* **38**, 514-527 (2013).
12. Lucas, M., Zhang, X., Prasanna, V. & Mosser, D.M. ERK activation following macrophage FcgammaR ligation leads to chromatin modifications at the IL-10 locus. *J Immunol* **175**, 469-477 (2005).
13. Weichhart, T., *et al.* The TSC-mTOR signaling pathway regulates the innate inflammatory response. *Immunity* **29**, 565-577 (2008).
14. Cheng, F., *et al.* A critical role for Stat3 signaling in immune tolerance. *Immunity* **19**, 425-436 (2003).
15. McBride, J.M., Jung, T., de Vries, J.E. & Aversa, G. IL-10 alters DC function via modulation of cell surface molecules resulting in impaired T-cell responses. *Cell Immunol* **215**, 162-172 (2002).
16. Cirone, M. EBV and KSHV Infection Dysregulates Autophagy to Optimize Viral Replication, Prevent Immune Recognition and Promote Tumorigenesis. *Viruses* **10**(2018).

17. Santarelli, R., *et al.* STAT3 activation by KSHV correlates with IL-10, IL-6 and IL-23 release and an autophagic block in dendritic cells. *Sci Rep* **4**, 4241 (2014).
18. You, L., *et al.* The role of STAT3 in autophagy. *Autophagy* **11**, 729-739 (2015).
19. Alissafi, T., *et al.* Tregs restrain dendritic cell autophagy to ameliorate autoimmunity. *J Clin Invest* **127**, 2789-2804 (2017).
20. Ghislat, G. & Lawrence, T. Autophagy in dendritic cells. *Cell Mol Immunol* **15**, 944-952 (2018).
21. Heckmann, B.L., Boada-Romero, E., Cunha, L.D., Magne, J. & Green, D.R. LC3-Associated Phagocytosis and Inflammation. *J Mol Biol* **429**, 3561-3576 (2017).
22. Hubbard-Lucey, V.M., *et al.* Autophagy gene Atg16L1 prevents lethal T cell alloreactivity mediated by dendritic cells. *Immunity* **41**, 579-591 (2014).
23. Hackstein, H., *et al.* Rapamycin inhibits IL-4--induced dendritic cell maturation in vitro and dendritic cell mobilization and function in vivo. *Blood* **101**, 4457-4463 (2003).
24. Xiong, A., *et al.* Flt3L combined with rapamycin promotes cardiac allograft tolerance by inducing regulatory dendritic cells and allograft autophagy in mice. *PLoS One* **7**, e46230 (2012).
25. Manouchehri, N., *et al.* Limitations of cell-lineage-specific non-dynamic gene recombination in CD11c.Cre(+)ITGA4(fl/fl) mice. *J Neuroimmunol* **344**, 577245 (2020).
26. Quinn, J.L., Kumar, G., Agasing, A., Ko, R.M. & Axtell, R.C. Role of TFH Cells in Promoting T Helper 17-Induced Neuroinflammation. *Front Immunol* **9**, 382 (2018).
27. Peng, Y., *et al.* Characterization of myelin oligodendrocyte glycoprotein (MOG)35-55-specific CD8+ T cells in experimental autoimmune encephalomyelitis. *Chin Med J (Engl)* **132**, 2934-2940 (2019).
28. Ji, Z., *et al.* Obesity Promotes EAE Through IL-6 and CCL-2-Mediated T Cells Infiltration. *Front Immunol* **10**, 1881 (2019).
29. Yoshimura, S., *et al.* IL-9 Controls Central Nervous System Autoimmunity by Suppressing GM-CSF Production. *J Immunol* **204**, 531-539 (2020).
30. Zhou, Y., *et al.* Tolerogenic dendritic cells induced by BD750 ameliorate proinflammatory T cell responses and experimental autoimmune encephalitis in mice. *Mol Med* **23**, 204-214 (2017).
31. Kalantari, T., *et al.* Tolerogenic dendritic cells produced by lentiviral-mediated CD40- and interleukin-23p19-specific shRNA can ameliorate experimental autoimmune encephalomyelitis by suppressing T helper type 17 cells. *Clin Exp Immunol* **176**, 180-189 (2014).
32. Melillo, J.A., *et al.* Dendritic cell (DC)-specific targeting reveals Stat3 as a negative regulator of DC function. *J Immunol* **184**, 2638-2645 (2010).
33. Weng, Q., *et al.* Lenalidomide regulates CNS autoimmunity by promoting M2 macrophages polarization. *Cell Death Dis* **9**, 251 (2018).

REVIEWER COMMENTS

Reviewer #1 (Remarks to the Author):

The newly revised manuscript has addressed my concerns and I think it is acceptable for publication in the NCOMMS.

Reviewer #3 (Remarks to the Author):

There are still some issues that need addressing in the revised manuscript:

- The new hypothesis is that Optn negatively regulates IL-10 expression through Jak2/Stat3, which leads to inhibition of DC maturation when Optn is lost. The idea that this is mediated by Stat3-dependent gene expression is based only on IL-10 gene. This should be supported by measuring other canonical Stat3-target genes, such as Socs3.

- Revised Fig.4D is missing total JAK and STAT3 loading controls.
Is the new qPCR data represented as mean of technical replicates or biological replicates?

- Interferon should be abbreviated to IFN not INF.

- The ChIP-seq data analysed in Fig.5 is from naive or IL-21 stimulated DC, which is illustrated in the genome browser images? If it is IL-21 stimulated, how is this relevant to LPS stimulation which is what it is being compared to?

- The experiments in Fig.5 using Optn-UBD mutant are not very convincing. The FACS shows only very small changes, what about IL-10 gene expression? The experiments are also not well controlled; expression levels of Optn/Optn-UBD mutant should be shown. An empty vector control should be used and a positive control for the the Optn-UBD mutant, for example inhibition of LPS-induced IFN β 1 expression.

- Gmcsf/IL4-derived BMDC are used throughout. This is a flawed system and contains many monocyte-derived cells that resemble CD11c+ activated macrophages (Helft et al. Immunity 2015). These limitations should be clearly stated and the justification for not using the more accepted FLT3L-derived BMDC system.

- Similarly, the use of CD11c-cre has the limitation of also targeting macrophages, as well as some other immune cells. So it is important to make this limitation clear and describe these mice as Optn-delta-CD11c or delta-Itgax, not delta-DC.

- Use of BMDM in vitro will not reflect CD11c+ macrophages in vivo, so this does not rule out similar effects of Optn deletion in CD11c+ macrophage populations. You could measure the genes measured in DC in CD11c+ macrophages in vivo to test this.

Point-to-point responses to Reviewers' concerns:

Reviewer #3 (Remarks to the Author):

There are still some issues that need addressing in the revised manuscript:

*1. The new hypothesis is that *Optn* negatively regulates *IL-10* expression through *Jak2/Stat3*, which leads to inhibition of DC maturation when *Optn* is lost. The idea that this is mediated by *Stat3*-dependent gene expression is based only on *IL-10* gene. This should be supported by measuring other canonical *Stat3*-target genes, such as *Socs3*.*

Response: We thank the reviewer for the suggestions. Our new hypothesis is that OPTN negatively regulates JAK2/STAT3 pathway, specifically IL-10-JAK2/STAT3-IL-10 looping is activated when *Optn* is lost, leading to the inhibition of DC maturation. And we did not rule out the role of any other STAT3 targets in the regulation of *Optn* deficient DC maturation. In this case, we appreciate other STAT3 targets that participate in DC maturation, such as SOCS3¹⁻⁴. ChIP-seq data confirmed that STAT3 could bind to *Socs3* gene in DCs (revised Supplementary Fig. 6), and our results showed the gene expression of *Socs3* was increased in *Optn* deficient BMDCs (revised Fig. 3d). SOCS3 is found to inhibit DC maturation together with subsequent Th17 response⁴. Thus, these data indicated that *Optn* deficiency could inhibit DC maturation at least partially via activating STAT3 target *Socs3*. We have included the new data in the revised Fig. 3d and Supplementary Fig. 6, and discussed this point in the revised text (page 12, line 3-8).

2. Revised Fig.4D is missing total JAK and STAT3 loading controls.

Is the new qPCR data represented as mean of technical replicates or biological replicates?

Response: We thank the reviewer for pointing this out. The missing total JAK2 and STAT3 loading controls were provided in revised Fig. 4d. And the new qPCR data shown in Fig. 5k is represented as mean of biological replicates, and we have plotted all the individual values in revised Fig. 5k.

3. Interferon should be abbreviated to IFN not INF.

Response: Thank you for the detailed comments. We have checked carefully and corrected the typos in the revised text and figures.

4. The ChIP-seq data analysed in Fig.5 is from naive or IL-21 stimulated DC, which is illustrated in the genome browser images? If it is IL-21 stimulated, how is this relevant to LPS stimulation which is what it is being compared to?

Response: Thank you for your critical comments. STAT3 can bind to chromatin when it is phosphorylated activated, and phosphorylated STAT3 is identified as a key format to manipulate STAT3 transcriptional function ^{5,6}. In our study, *Optn* deficiency in BMDCs stimulated by LPS increased STAT3 phosphorylation specifically (revised Fig. 3a-c). In the referenced paper, IL-21 was also found to upregulate STAT3 phosphorylation and even have a synergistic effect with LPS in DCs ⁷⁻¹⁰. Following tyrosine phosphorylation and subsequent translocation to the nucleus, p-STAT3 binds to promoters containing the consensus sequences TT(N4)AA or TT(N5)AA ¹¹, which means increased p-STAT3 after different stimuli should have similar binding gene profiles. Thus, what we look for is the binding gene profile of p-STAT3 in DCs. In this case, we assume that IL-21 stimulation is relevant to LPS stimulation in DCs since both the stimuli activate STAT3 phosphorylation to bind its target genes.

5. The experiments in Fig.5 using *Optn-UBD* mutant are not very convincing. The FACS shows only very small changes, what about *IL-10* gene expression? The experiments are also not well controlled; expression levels of *Optn/Optn-UBD* mutant should be shown. An empty vector control should be used and a positive control for the the *Optn-UBD* mutant, for example inhibition of LPS-induced *IFN β 1* expression.

Response: Thank you for your critical suggestions. We agree that the FACS data in Supplementary Fig. 3f, g showed small changes with about 10% increase of CD80⁺ or CD86⁺ cells after OPTN or OPTN ^{Δ UBD} overexpression. However, it still makes sense that a change of 10% is sufficient to have a significant impact on DC maturation and function ¹²⁻¹⁴. As expected, both OPTN and OPTN ^{Δ UBD} overexpression also restored the gene expression of *Il-10* in LPS-stimulated *Optn* deficient BMDCs (revised Supplementary Fig. 3h), indicating that the function of OPTN on STAT3-dependent *Il-10* expression is also independent of its ubiquitin-binding domain.

Besides, we have included the expression levels of OPTN and OPTN ^{Δ UBD} (revised Supplementary Fig. 3e). We have included the empty vector control in the study actually, and we changed the marker “LPS” to “LPS + pCDH” to show the empty vector control (revised Supplementary Fig. 3e-h). We have discussed this point in the revised text (page 8, line 18-24).

6. *Gmcsf/IL4*-derived BMDC are used throughout. This is a flawed system and contains many monocyte-derived cells that resemble *CD11c+* activated macrophages (Helft et al.

Immunity 2015). These limitations should be clearly stated and the justification for not using the more accepted FLT3L-derived BMDC system.

Response: Thank you for the critical comments. GM-CSF/IL-4 is a widely used system for bone marrow derived dendritic cells culture. As shown by *Helft, et al*, application of GM-CSF (20 ng/ml) resulted in 63.4 % CD11c⁺MHC-II⁻CD11b^{high} cells with 90.7 % CD115⁺ macrophages in this cell group, and the percentage of which dropped to 32.1 % and 74.8 % when added IL-4 (5 ng/ml) to GM-CSF culture system¹⁵. Based on this study, we used GM-CSF (50 ng/ml) and IL-4 (20 ng/ml) for BMDCs culture, and macrophage differentiation could be inhibited due to high concentration of IL-4. As expected, only 17.0 % CD11c⁺MHC-II⁻CD11b^{high} cells were counted, and the CD115⁺ macrophages were almost absent in both CD11c⁺MHC-II⁻CD11b^{high} and CD11c⁺MHC-II⁺CD11b^{int} cells (revised Supplementary Fig. 1). Taken together, GM-CSF (50 ng/ml) and IL-4 (20 ng/ml) system is appropriate for BMDCs culture. We also appreciate the FLT3L-derived BMDCs system, and both of them can be used for BMDCs culture. We have included the new data in the revised Supplementary Fig. 1, and discussed this point in the revised text (page 4, line 28 to page 5, line 2 and page 13, line 29 to page 14, line 6).

7. Similarly, the use of CD11c-cre has the limitation of also targeting macrophages, as well as some other immune cells. So it is important to make this limitation clear and describe these mice as Optn-delta-CD11c or delta-Itgax, not delta-DC.

Response: Thank you for your critical comments. We agree that *CD11c-cre* can also target CD11c⁺ macrophages and a minor subpopulation of lymphocyte like T-cell¹⁷, and we have replaced all *Optn*^{ADC} mice into *Optn*^{ACD11c} mice and discussed the use of *CD11c-cre* mice in the revised manuscript (page 5, line 8-9 and page 14, line 7-17).

8. Use of BMDM in vitro will not reflect CD11c+ macrophages in vivo, so this does not rule out similar effects of Optn deletion in CD11c+ macrophage populations. You could measure the genes measured in DC in CD11c+ macrophages in vivo to test this.

Response: We thank the reviewer for the suggestions. CD11c⁺ macrophages are identified as activated, proinflammatory or M1-like macrophages¹⁸⁻²⁰, and a portion of CD11c⁺ macrophages *in vivo* can be derived from bone marrow or monocyte²⁰⁻²³. As expected, we found that comparable numbers of CD11c⁺ BMDMs from ctrl and *Optn*^{ACD11c} mice (revised Supplementary Fig. 5f). Moreover, similar percentage of pro-inflammatory F4/80⁺CD11c⁺CD86⁺ BMDMs and anti-inflammatory F4/80⁺CD11c⁺CD206⁺ BMDMs from ctrl and *Optn*^{ACD11c} mice were detected after LPS or IL-4 stimulation (revised Supplementary Fig. 5g, h). Together, these data may rule out the partial effects of *Optn*

deletion in CD11c⁺ macrophages. We have included new data in revised Supplementary Fig. 5, and discussed this point in the revised text (page 9, line 22-25 and page 14, line 10-17).

Reference:

1. Auernhammer, C.J., Bousquet, C. & Melmed, S. Autoregulation of pituitary corticotroph SOCS-3 expression: characterization of the murine SOCS-3 promoter. *Proc Natl Acad Sci U S A* **96**, 6964-6969 (1999).
2. Baker, B.J., Qin, H. & Benveniste, E.N. Molecular basis of oncostatin M-induced SOCS-3 expression in astrocytes. *Glia* **56**, 1250-1262 (2008).
3. Kim, Y.D., *et al.* Metformin ameliorates IL-6-induced hepatic insulin resistance via induction of orphan nuclear receptor small heterodimer partner (SHP) in mouse models. *Diabetologia* **55**, 1482-1494 (2012).
4. Shi, D., *et al.* SOCS3 ablation enhances DC-derived Th17 immune response against *Candida albicans* by activating IL-6/STAT3 in vitro. *Life Sci* **222**, 183-194 (2019).
5. Shuai, K. & Liu, B. Regulation of JAK-STAT signalling in the immune system. *Nat Rev Immunol* **3**, 900-911 (2003).
6. Hillmer, E.J., Zhang, H., Li, H.S. & Watowich, S.S. STAT3 signaling in immunity. *Cytokine Growth Factor Rev* **31**, 1-15 (2016).
7. Jian, L., *et al.* Interleukin-21 enhances Toll-like receptor 2/4-mediated cytokine production via phosphorylation in the STAT3, Akt and p38 MAPK signalling pathways in human monocytic THP-1 cells. *Scand J Immunol* **89**, e12761 (2019).
8. Yang, Z., Wu, C.M., Targ, S. & Allen, C.D.C. IL-21 is a broad negative regulator of IgE class switch recombination in mouse and human B cells. *J Exp Med* **217**(2020).
9. Vallieres, F. & Girard, D. Mechanism involved in interleukin-21-induced phagocytosis in human monocytes and macrophages. *Clin Exp Immunol* **187**, 294-303 (2017).
10. van der Fits, L., Out-Luiting, J.J., Tensen, C.P., Zoutman, W.H. & Vermeer, M.H. Exploring the IL-21-STAT3 axis as therapeutic target for Sezary syndrome. *J Invest Dermatol* **134**, 2639-2647 (2014).
11. Horvath, C.M., Wen, Z. & Darnell, J.E., Jr. A STAT protein domain that determines DNA sequence recognition suggests a novel DNA-binding domain. *Genes Dev* **9**, 984-994 (1995).
12. Michalski, J., *et al.* Quercetin induces an immunoregulatory phenotype in maturing human dendritic cells. *Immunobiology* **225**, 151929 (2020).
13. Li, G., *et al.* Effects of combined treatment with PDL1 Ig and CD40L mAb on immune tolerance in the CBA/J x DBA/2 mouse model. *Mol Med Rep* **21**, 1789-1798 (2020).

14. Li, J.G., *et al.* CD80 and CD86 knockdown in dendritic cells regulates Th1/Th2 cytokine production in asthmatic mice. *Exp Ther Med* **11**, 878-884 (2016).
15. Helft, J., *et al.* GM-CSF Mouse Bone Marrow Cultures Comprise a Heterogeneous Population of CD11c(+)MHCII(+) Macrophages and Dendritic Cells. *Immunity* **42**, 1197-1211 (2015).
16. Son, Y.I., *et al.* A novel bulk-culture method for generating mature dendritic cells from mouse bone marrow cells. *J Immunol Methods* **262**, 145-157 (2002).
17. Manouchehri, N., *et al.* Limitations of cell-lineage-specific non-dynamic gene recombination in CD11c.Cre(+)ITGA4(fl/fl) mice. *J Neuroimmunol* **344**, 577245 (2020).
18. Zeng, J., *et al.* Stearic Acid Induces CD11c Expression in Proinflammatory Macrophages via Epidermal Fatty Acid Binding Protein. *J Immunol* **200**, 3407-3419 (2018).
19. Ono, Y., *et al.* CD11c+ M1-like macrophages (MPhis) but not CD206+ M2-like MPhi are involved in folliculogenesis in mice ovary. *Sci Rep* **8**, 8171 (2018).
20. Wang, S., *et al.* Luteolin transforms the polarity of bone marrow-derived macrophages to regulate the cytokine storm. *J Inflamm (Lond)* **18**, 21 (2021).
21. Yadav, S., Pandey, S.K., Goel, Y., Temre, M.K. & Singh, S.M. Antimetabolic Agent 3-Bromopyruvate Exerts Myeloprotentiating Action in a Murine Host Bearing a Progressively Growing Ascitic Thymoma. *Immunol Invest* **49**, 425-442 (2020).
22. Meester, I., Rosas-Taraco, A.G. & Salinas-Carmona, M.C. *Nocardia brasiliensis* induces formation of foamy macrophages and dendritic cells in vitro and in vivo. *PLoS One* **9**, e100064 (2014).
23. Vishvakarma, N.K. & Singh, S.M. Augmentation of myelopoiesis in a murine host bearing a T cell lymphoma following in vivo administration of proton pump inhibitor pantoprazole. *Biochimie* **93**, 1786-1796 (2011).